# High resolution assessment of commercial fisheries activity along the US West Coast using Vessel Monitoring System data with a case study using California groundfish fisheries

Yi-Hui Wang[1¤], Benjamin I. Ruttenberg[1,2]*, Ryan K. Walter[1,3], Frank Pendleton[4], Jameal F. Samhouri[5], Owen R. Liu[5], Crow White[1,2]

1 Center for Coastal Marine Sciences, California Polytechnic State University, San Luis Obispo, California, United States of America, 2 Biological Sciences Department, California Polytechnic State University, San Luis Obispo, California, United States of America, 3 Physics Department, California Polytechnic State University, San Luis Obispo, California, United States of America, 4 Bureau of Ocean Energy Management, Camarillo, California, United States of America, 5 National Oceanic and Atmospheric Administration, National Marine Fisheries Service, Northwest Fisheries Science Center, Seattle, Washington, United States of America

¤ Current address: Ocean Protection Council, California Natural Resources Agency, Sacramento, California, United States of America

* bruttenb@calpoly.edu

**Data Availability Statement:** Data are available at the NOAA National Centers for Environmental

## Abstract

Commercial fisheries along the US West Coast are important components of local and regional economies. They use various fishing gear, target a high diversity of species, and are highly spatially heterogeneous, making it challenging to generate a synoptic picture of fisheries activity in the region. Still, understanding the spatial and temporal dynamics of US West Coast fisheries is critical to meet the US legal mandate to manage fisheries sustainably and to better coordinate activities among a growing number of users of ocean space, including offshore renewable energy, aquaculture, shipping, and interactions with habitats and key non-fishery species such as seabirds and marine mammals. We analyzed vessel tracking data from Vessel Monitoring System (VMS) from 2010 to 2017 to generate high-resolution spatio-temporal estimates of contemporary fishing effort across a wide range of commercial fisheries along the entire US West Coast. We identified over 247,000 fishing trips across the entire VMS data, covering over 25 different fisheries. We validated the spatial accuracy of our analyses using independent estimates of spatial groundfish fisheries effort generated through the NOAA's National Marine Fisheries Service Observer Program. Additionally, for commercial groundfish fisheries operating in federal waters in California, we combined the VMS data with landings and ex-vessel value data from California commercial fisheries landings receipts to generate highly resolved estimates of landings and ex-vessel value, matching over 38,000 fish tickets with VMS data that included 87% of the landings and 76% of the ex-vessel value for groundfish. We highlight fisheries-specific and spatially-resolved patterns of effort, landings, and ex-vessel value, a bimodal distribution of fishing

Information, Accession #0291013, at: https://www.ncei.noaa.gov/archive/accession/0291013.

**Funding:** This work was funded by the US Department of the Interior, Bureau of Ocean Energy Management (BOEM), Environmental Studies Program, Washington, DC (Agreement Number #M16AC00023, https://www.boem.gov/) and the California Ocean Protection Council (OPC) (Agreement #C021040, https://www.opc.ca.gov/) to BR, CW, and RW. BOEM and OPC had a role in reviewing the manuscript. FP at BOEM assisted in manuscript preparation.

**Competing interests:** The authors have declared that no competing interests exist.

effort with respect to depth, and variable and generally declining effort over eight years. The information generated by our study can help inform future sustainable spatial fisheries management and other activities in the marine environment including offshore renewable energy planning.

## 1. Introduction

Commercial fisheries along the US West Coast are diverse and productive [1], generating economic benefits for local and regional economies and working waterfront communities [2]. On the US West Coast, commercial fisheries employ over 2,800 vessel owners and a much larger number of crew [3], and in California generate hundreds of millions of dollars of ex-vessel revenues annually [4]. The potential socio-economic impacts of fishery management decisions have led to a considerable research focus on assessing the spatial and temporal patterns in fishing effort, fish stocks, and fisheries landings and ex-vessel value.

Understanding the spatial distribution of activity across US West Coast fisheries is becoming more important to ensure fisheries sustainability as other ocean uses increase and regulatory and management approaches increase in complexity. Vessel traffic for shipping has been increasing in recent years, and planning for offshore wind energy and marine aquaculture is moving forward on the West Coast [5–7]. Networks of marine protected areas have been established in California, Oregon, and Washington over the last 12 years [8–10], and the size and shape of Rockfish Conservation Areas that span the US West Coast have changed multiple times in the last decade and are likely to change again as stocks recover [11]. Furthermore, commercial fisheries are responding dynamically to changes in the marine environment driven by climate change [12–14]. To support improved fisheries management and support decision making for multiple uses of ocean space along the US West Coast in a changing marine environment, more accurate and precise data are needed on current spatio-temporal fisheries dynamics across the region.

Recent analyses of California commercial fisheries landings receipts ("fish tickets") at relatively coarse grid-block spatial resolution (the highest spatial resolution of 10' x 10' blocks, approximately 15 km zonally and 18.5 km meridionally depending on latitude) has revealed some important patterns and trends. There has been a shift from high-volume, low-value species toward low-volume, high-value species [15, 16], along with changes in regional spatial patterns of fishing activity over the last decade [16]. While useful for broad-scale analyses, fish ticket data are limited in their application. First, the spatial resolution of fish ticket data is coarse, resulting in highly imprecise landings data in many fisheries [16]. Second, self-reported block information that accompanies commercial fish tickets may be inaccurate—for example, reporting blocks that are unrealistic because they are outside the biological depth limit of the target species [16, 17]. Finally, fish ticket data report landings and ex-vessel value (i.e., revenue), but lack reliable spatially-explicit estimates of fishing effort, reducing their utility in assessing fishing activity. As a result, fish ticket data are limited on their own in fishery management and marine spatial planning applications that require more precise information about fishing activity.

Analyses of vessel monitoring system (VMS) data have been increasingly used to explore spatial fishing activity and inform fisheries management (e.g., [18–22]). VMS provides near real-time transmission of the spatial position of a specific vessel at discrete time intervals (typically every hour in the U.S.), allowing for highly-accurate monitoring of fishing vessel

movements. While VMS does not explicitly differentiate between fishing activity and non-fishing activities (e.g., transit), these activities can be distinguished in relation to vessel speed calculated from consecutive VMS points (see e.g., [21, 23, 24]). Moreover, VMS provides high spatial accuracy of vessel location (~100 m spatial resolution), greatly reduces misreporting of fishing location, and enables quantification of fishing effort [25].

While there is a growing use of VMS data in fisheries science and management [18, 21, 26, 27], these data present challenges of their own. Most importantly, VMS data do not provide fisheries landings or ex-vessel value estimates, nor is VMS required for all vessels and fisheries. However, for fisheries and fleets where VMS is widely adopted and/or required by law, VMS data can provide reliable and representative spatial estimates of fishing effort. Further, VMS data contains a unique vessel identification number for a specific vessel, which can be matched with vessel identification number and date-stamp data in fish tickets, thereby allowing the two datasets to be linked [20, 28, 29]. Fish ticket landings and ex-vessel value reports can then be mapped onto VMS vessel tracks to generate accurate and spatially resolved estimates of landings and ex-vessel value that cannot be provided by fish ticket or VMS data alone.

In this study, we focused on VMS data rather than Logbook and Observer data since VMS data are relatively underexplored. We processed the data, and produced spatial maps of relative fishing effort across the entire US West Coast, encompassing, a variety of commercial fisheries that collected VMS data (see Data and Methods). We then compared VMS data with independent Observer data and Logbook data and found similar spatial patterns that suggest that our analysis on VMS data can reasonably capture fishing activity patterns. Because VMS is required for all vessels fishing in US federal waters for groundfish [30], as a test case we conducted a more comprehensive analysis of the spatio-temporal dynamics of groundfish in particular by matching VMS polls in California with concurrent California fish ticket landings and ex-vessel value data for groundfish. The resulting VMS-fish ticket dataset enabled us to assess patterns of effort, landings and ex-vessel value at high resolution along the entire California coast. The data synthesis framework demonstrated here for the groundfish fishery using independent tracking (VMS) and reporting (fish ticket) data can be used for a variety of marine spatial planning and fishery management applications for groundfish, and our framework, which builds upon previous research using VMS data to quantify fisheries activities [18, 21, 26, 27 and others], can be applied to other fisheries to generate similar information on spatial fisheries dynamics.

## 2. Data and methods

### 2.1 Data

**Vessel Monitoring System.** We obtained VMS data from National Oceanic and Atmospheric Administration's (NOAA) National Marine Fisheries Service (NMFS) Office of Law Enforcement (OLE) [25] from 2010 to 2017 to estimate fishing effort across the entire US West Coast within the US Exclusive Economic Zone (EEZ, 200 nautical miles [nm] from the coast). VMS is a spatial monitoring system required on certain fishing vessels that uses satellite and cellular-based communications from onboard transceiver units to track the geographic location of US commercial fishing vessels. The transceiver units send 'polls' (also referred to as 'pings' and 'records'), which are transmissions of data that include vessel identification, time, date, location, and average speed [31]. VMS data tracks vessel locations at high temporal frequency [32]; of the VMS data used in this study, over 87% of the polls had transmission intervals less than or equal to one hour.

On the US West Coast, VMS transceivers are required for commercial fishing vessels registered with a Pacific Coast groundfish limited entry permit, a fishing permit issued by NOAA

NMFS. This includes vessels that take, retain, or possess groundfish within federal waters in the EEZ, or land groundfish taken in the EEZ [32]. 'Groundfish' includes all benthically associated fishery species on the west coast, such as sablefish, flatfish (e.g., Dover sole), and rockfishes [15, 16]. In addition, VMS is required on drift gillnet vessels participating in Highly Migratory Species (HMS) fisheries, and any vessel that uses non-groundfish trawl gear in the EEZ [32]. Although the VMS system is associated with permitted fishing in federal waters, vessels regulated to have VMS are required to operate the transceiver during all fishing trips, whether in federal or state waters. Note that vessels that fish for groundfish only in state waters (out to 3 nautical miles from shore) are not required to have VMS. In addition, vessels that operate under a permit associated with VMS (e.g., Pacific Coast groundfish limited entry permit) are also required to operate a VMS transceiver during all fishing trips, even when fishing with gear or targeting fish species outside the jurisdiction of the VMS requirement (e.g., Dungeness crab fishery, salmon fishery). In addition to tracking information, the VMS polls include a declaration code characterizing the fishing activity, including gear and target species, which are generally associated with a specific fishing permit (Table 1). Thus, VMS not only has nearly complete coverage of commercial groundfish fisheries operating in federal waters, but also covers many other commercial fisheries in the region (albeit with varying, potentially low, levels of representation). Because the groundfish fisheries have high coverage, we matched its VMS data with the fish ticket data for groundfish landings to create high-resolution maps of landings and ex-vessel value.

**Fish ticket.** We used commercial fisheries landings receipts, known as 'fish tickets,' which offer information about landings and ex-vessel values not provided by VMS data. The fish ticket data in this study focuses on California, as it was provided by California Department of Fish and Wildlife (CDFW) through a data agreement over the same time period as the VMS data. Fish tickets are submitted to CDFW at port following a vessel fishing trip. Along with a unique vessel identification number, each ticket records the landing weight and unit price (i.e., price per pound), species caught, landing port and date, and fishing block catch location (see [16]).

## 2.2 Data processing

**Vessel Monitoring System.** We used raw VMS data for the years 2010–2017. We applied a series of processing and filtering steps to the raw VMS data (Fig 1). We first corrected negative latitudes of VMS polls due to transmission errors to positive latitudes (~0.39% of the raw data). We then removed duplicate VMS polls so that there was only one VMS poll at a given time for a vessel (~9.5% of the raw data). To focus on activities along the US West Coast, we removed VMS polls from outside the domain of 32-50˚N and 117-130˚W, leaving ~78.4% of the raw data retained. Of all VMS polls outside the West Coast domain, 86.9% were in the vicinity of Alaska (within the domain of 50-70˚N and 130-180˚W), 2.8% were near Hawaii (within the domain of 18-23˚N and 154.5-161˚W) and most of the remaining VMS polls were distributed across the Pacific. The bulk of the VMS polls in the remaining dataset were from vessels docked in port. A VMS poll was labelled as out of port if it was not on the land of the West Coast states (i.e., California, Oregon, and Washington) and was outside of a 3-km buffer zone of ports for commercial fisheries (~12.5% of the raw data retained for downstream analysis, which includes the calculation of fishing efforts by declarations and the match with fish ticket data for groundfish fisheries). We filtered VMS polls within 3 km of a port as an educated guess to minimize the chance of misidentifying polls from vessels moving slowly near ports as fishing. The VMS data loss in each processing and filtering step is visualized in S1 Fig.

**Table 1. Information summary for VMS declaration codes.**

| Decl. | Description (dominant gear types) | Number of VMS records | Number of vessels | Cutoff fishing speed (knots) | | % of fishing effort inside biological depth limit |
|---|---|---|---|---|---|---|
| | | | | local min | slope min | |
| 210* | Limited entry fixed gear not including shorebased IFQ (pots/traps, bottom longline, longline, hook and line) | 547,116 | 266 | 4.51 | 2.39 | 97.4% |
| 211* | Limited entry groundfish non-trawl shorebased IFQ (pots/traps, bottom longline, longline, hook and line) | 111,976 | 47 | 4.97 | 2.01 | 99.2% |
| 220* | Limited entry midwater trawl gear non-whiting shorebased IFQ (midwater trawl for species other than whiting) | 85,600 | 54 | 4.94 | 3.63 | NDL |
| 221* | Limited entry midwater trawl Pacific whiting shorebased IFQ (midwater trawl for whiting) | 316,994 | 35 | 4.31 | 3.27 | NDL |
| 222* | Limited entry midwater trawl Pacific whiting catcher-processor sector (midwater trawl for whiting) | 174,298 | 16 | 5.48 | 4.13 | NDL |
| 223* | Limited entry midwater trawl Pacific whiting mothership sector; catcher vessel or mothership (midwater trawl for whiting) | 234,673 | 32 | 6.00 | 3.40 | NDL |
| 230* | Limited entry bottom trawl shorebased IFQ not including demersal trawl | 635,890 | 129 | 4.41 | 2.46 | ~100% |
| 231* | Limited entry demersal trawl shorebased IFQ | 7,745 | 2 | 4.34 | 2.01 | 100% |
| 233* | Open access longline gear for groundfish | 186,528 | 414 | 4.03 | 2.01 | 97.3% |
| 234* | Open access groundfish trap or pot gear | 136,693 | 289 | 4.00 | 2.01 | 98.9% |
| 235* | Open access line gear for groundfish | 260,969 | 307 | 4.24 | 2.00 | 94.0% |
| 240 | Non-groundfish trawl gear for ridgeback prawn | 21,527 | 4 | 4.63 | 3.00 | 99.6% |
| 241 | Non-groundfish trawl gear for pink shrimp | 864,053 | 141 | 4.61 | 2.01 | 98.3% |
| 242* | Non-groundfish trawl gear for California halibut | 44,187 | 22 | 4.95 | 2.92 | 99.6% |
| 243 | Non-groundfish trawl gear for sea cucumber | 51,849 | 11 | 5.23 | 2.10 | 89.7% |
| 250* | Tribal trawl gear | 3,156 | 4 | 6.00 | 3.90 | 100% |
| 260 | Open access prawn trap or pot gear | 37,153 | 21 | 4.92 | 2.70 | 96.8% |
| 261 | Open access Dungeness crab trap or pot gear | 1,211,912 | 579 | 4.93 | 3.11 | 95.3% |
| 262* | Open access Pacific Halibut longline gear | 24,836 | 124 | 3.64 | 2.02 | 96.1% |
| 263 | Open access salmon troll gear | 571,713 | 384 | 4.40 | 3.01 | NDL |
| 264* | Open access California halibut line gear | 37,738 | 47 | 4.47 | 2.53 | 93.7% |
| 265* | Open access sheephead trap or pot gear | 9,802 | 8 | 6.00 | 2.14 | 73.8% |
| 266 | Open access Highly Migratory Species line gear | 473,212 | 344 | 3.74 | 6.00 | NDL |
| 267 | Open access Coastal Pelagic Species net gear | 2,277 | 8 | 5.17 | 2.84 | NDL |
| 268 | Open access California gillnet complex gear | 32,966 | 15 | 6.00 | 2.19 | NDL |
| 269 | A gear that is not listed above | 408,909 | 495 | 4.55 | 3.81 | NDL |

For each VMS declaration code, this table includes the description, number of VMS polls after data processing used in this study, number of vessels represented, fishing-transit cutoff speed determined from the local minimum method, fishing-transit cutoff speed determined from the slope minimum method, and the percentage of fishing effort inside the maximum depth limit relative to the total fishing effort over lease blocks based on the local minimum method (NDL represents declarations whose fisheries have no depth limits).

An asterisk (*) indicates a declaration for the groundfish fisheries that have VMS requirements.

Additional information on VMS data and the declaration codes is provided by NOAA at <https://www.fisheries.noaa.gov/west-coast/resources-fishing/vessel-monitoring-system-west-coast>

For each vessel, we calculated the average speed at each poll location based on the time and distance between the current poll and previous poll, reflecting the speed a vessel has been traveling over some distance. Our calculated average speed agrees well with the average speed directly provided with the VMS polls and has fewer extremely large outliers than the original

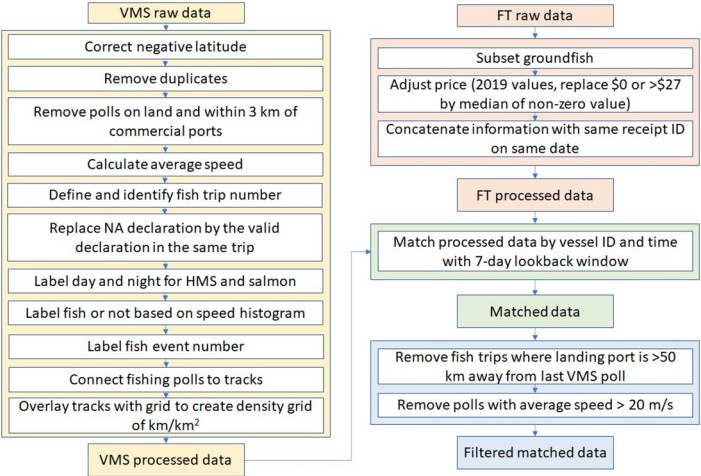

**Fig 1. Flowchart illustrating the processing of VMS and fish ticket data.**

average speed. Instantaneous vessel speed is provided by VMS, but we did not use it because reported speed is more susceptible to transient fluctuations in vessel speed and reported speed was missing for many polls. Hereafter, 'speed' refers to the calculated average vessel speed.

Next, we calculated individual trips for a given vessel, with the beginning of a trip defined as when that vessel left a port and the end of the trip when the same vessel returned to a port. Therefore, our analysis included all vessels that returned to the same port or to a different port. All consecutive polls for a given vessel between the start and end of a trip were considered to be part of a single trip, unless two consecutive polls of a vessel were recorded more than five hours apart. In that case, a new trip was defined after the time gap since it is impossible to know where that vessel was during that 5+ hour gap in VMS coverage. With only roughly 0.21% of VMS polls with 5+ hour gaps, the likelihood of these time gaps substantially altering the results is extremely low.

To organize the data by fisheries type, we grouped VMS polls by declaration code (Table 1). Approximately 9.83% of the processed polls had invalid or missing declaration codes, which typically arose from entire trips not having any declaration data. These VMS polls were not excluded from matching with fish tickets if other information is available. While a vessel can switch declaration codes on a given trip, less than 1% of the trips recorded more than one valid declaration code.

To estimate fishing versus non-fishing activity, we generated density histograms of vessel speed for all out-of-port VMS polls for each declaration code [33, 34]. Many of the declarations had bimodal distributions of speed. We assumed vessels transit to/from fishing grounds at a relatively higher cruising speed than when fishing at the fishing grounds, thus in the histograms the lower mode represented the peak of intensity of speeds while fishing, and the higher mode the peak of intensity of speeds while transiting. Additionally, we estimated the local minimum between the modes in the density histograms as the cutoff speed, with locations with speeds lower than this cutoff classified as fishing and locations with speeds greater than or equal to this cutoff categorized as transiting. We constrained the local minimum between two and six knots (3.7–11.1 km/hr), given that previous research found typical fishing speeds for US commercial fisheries vessels to be within or below this range [33, 35].

To investigate the sensitivity of our results to the method of using the local minimum vessel speed for differentiating between fishing and non-fishing activity, we compared the spatial

pattern of fishing effort for each declaration code derived from the local minimum speed method with that derived from an alternative method where we used the minimum slope of the density histogram between two and six knots (i.e., the point of steepest drop, or most negative slope). See S2–S4 Figs for an illustration of the two methods.

For salmon and HMS, we further classified VMS polls as 'daytime' if they occurred between morning and evening nautical twilight and labelled all polls outside this time as 'not fishing' since these two fisheries target visual predators and therefore typically only operate during daytime. Finally, we define a 'fishing event' within each trip as a series of consecutive polls that were classified as 'fishing'. The resulting track of each fishing event was used to quantify fishing effort for each declaration code (Section 2.3).

**Fish tickets.**   We retained fish tickets for groundfish, since VMS is required for groundfish fisheries in federal waters. The fish tickets pertaining to groundfish species, including leopard shark, sanddabs, founders (starry, unspecified), turbots, soles (bigmouth, rock, fantail, sand, English, butter, tongue), California halibut, cabezon, scorpionfish, staghorn sculpin, yellowfin sculpin, lingcod, petrale sole, Pacific halibut, North Pacific hake, flounder arrowtooth, rockfish, sablefish, spiny dogfish, soles (Dover, rex) and thornyheads, were extracted from the fish ticket raw data (see detailed information in S1 Table of Wang et al. 2022). Before matching with VMS data, we processed fish ticket data following [16]. We corrected unit prices for inflation based on the consumer price index for December 2019. We also replaced null values and values of $0/lb, taking into consideration the presumed economic value of landings where values were not reported, or implausible unit prices (> $27/lb), with the median price of non-$0 prices for that species or fishery group. We then calculated the ex-vessel value, or revenue, of the catch landed by multiplying the landing weight in pounds by the unit price reported on each fish ticket. If multiple fish tickets were submitted for a given vessel on the same date, these data were combined.

For visualization of patterns of landings and ex-vessel value by groundfish species, we aggregated the species recorded in fish tickets into either individual species or species functional groups. These seven categories included: halibut (California halibut, *Paralichthys californicus*, and Pacific halibut, *Hippoglossus stenolepis*), flatfish (all other flatfish excluding halibut), roundfish (i.e., cabezon, *Scorpaenichthys marmoratus*, lingcod, *Ophiodon elongatus*, and scorpionfish, *Scorpaena sp.*), rockfish (*Sebastes sp.*), sablefish (*Anoplopoma fimbria*), thornyheads (*Sebastolobus sp.*), and other (i.e., spiny dogfish sharks, *Squalidae acanthias.*). We defined functional groups following PacFIN (https://pacfin.psmfc.org/pacfin_pub/data_rpts_pub/code_lists/sp.txt) with a few modifications. Halibut was evaluated separately from the flatfish category because these species are specifically targeted and represented by specific declaration codes not associated with the other flatfishes (Table 1). Sablefish was separated because of its relatively high commercial ex-vessel value in the study region.

## 2.3 Fishing effort

Using the processed VMS data, we quantified fishing effort for each declaration code across lease blocks designated by the Bureau of Ocean Energy Management (BOEM). BOEM lease blocks are 4.8 x 4.8 km polygons covering the EEZ adjacent to California, Oregon, and Washington [36]. To calculate fishing effort, for each declaration code we connected the VMS polls within each fishing trip chronologically to create a set of fishing tracks, each corresponding with a fishing event, then we summed the length of the fishing tracks inside each lease block. Following [26], we then divided the sum of the fishing track lengths within each lease block by the area of the block, then averaged the solution across the number of years in the data set (eight) to determine the annual-mean fishing effort in each lease block (km/km$^2$/year). To

assess the spatial reliability of the VMS data, for each declaration code we calculated the percentage of estimated fishing effort across lease blocks within versus beyond the biological depth limit for the target species (delineated by [15, 37]), similar to that conducted by [16].

To help validate the spatial accuracy of our results, we compared the estimates of the spatial distribution of fisheries effort generated in this study to estimates of effort generated through the NOAA NMFS Observer Program [26], which uses independent scientific observation of fishing activity and catch on commercial groundfish vessels while at sea. Coverage rates for groundfish trawl fisheries (from 2011-present) in the Observer Program are extremely high, generally >99% [38]. Therefore, the Observer Program dataset provides a robust 'ground-truth' for estimates of effort derived from VMS data. We focused on groundfish trawl fisheries (primarily represented by Declaration Code 230) in the VMS dataset, since this declaration code is the primary fishery represented in the NOAA groundfish trawl Observer dataset [26]. We summarized VMS data by years 2011–2015 and 2016–2017 to match similar summary periods in [26]. We downloaded these Observer datasets from Data Basin (https://databasin.org/datasets/8b0d742d072746cca3bb98be0c9c49d8/) and re-gridded fishing effort at the same BOEM lease block scale used for our analyses to enable direct comparisons between these two datasets. Finally, we performed a spatial correlation analysis on fishing effort over the lease blocks that were available in both the VMS and Observer datasets.

Importantly, the coverage rate of the VMS and Observer data differs between fisheries [38]. While the VMS covers nearly all trips for groundfish fisheries (because it is required—see e.g., [32]), it likely covers less for other fisheries. We are not aware of any source that explicitly quantifies coverage rate for US West Coast fisheries that utilize VMS.As a result, VMS data may not capture the absolute magnitude of fishing activity for many of the non-groundfish fisheries for which VMS is not required. There are few other datasets with which to ground truth our VMS results, but a map using the Oregon Department of Fish and Wildlife Logbook data made available to the OROWindMap gateway (https://offshorewind.westcoastoceans.org/)https://offshorewind.westcoastoceans.org/) for the Dungeness crab fishery for 2010/2011-2017/18 nearly completely overlaps with our VMS dataset. While we were unable to download the spatial data—and therefore could not statistically compare it to our data—we used this dataset to compare the visual patterns for this fishery.

## 2.4 Matching VMS and fish ticket data for groundfish

To link the processed VMS data (vessel records of fishing vs. non-fishing activity across declaration codes) with the California fish ticket data (port landings and ex-vessel value reported by vessels), we followed methodology developed by NOAA [20, 28, 35], summarized here. The two data sets were first joined by unique vessel identification numbers and time. If VMS polls were recorded between two consecutive fish tickets for a given vessel, then those polls were matched to the later fish ticket. If there were more than seven days between two fish tickets, only the VMS polls for the last seven days were included matched to the later fish ticket. We chose a seven-day interval as the cutoff interval between consecutive fish tickets because the frequency distribution of intervals between consecutive fish tickets showed a substantial decline after seven days. If the reported port of landing from the fish ticket was more than 50 km away from the last VMS poll of an associated trip, all VMS polls from that trip were removed from further analysis. Finally, if a VMS poll recorded a speed greater than 20 m/s (~39 knots), it was removed from further analysis. Importantly, we considered only vessel name, location, and date—but not declaration code—when matching vessel trips to fish tickets, so potentially erroneous declaration codes had no influence on the matching process. See

Fig 1 for a graphical description of the data processing steps for VMS and fish ticket data, including the process to join these datasets.

## 2.5 Fishery landings and ex-vessel value

To estimate the spatial distribution of landings and ex-vessel value, we assigned the landings and value from a given fish ticket to the fishing tracks determined from its matched VMS polls, then distributed landings and value proportionally within each fishing block based on the fraction of the track length that occurred in each block for that fish ticket. In some cases, fish tickets matched multiple VMS trips (22% of fish tickets) so catch for that fish ticket was distributed proportionally over the length of the total fishing tracks for those matching trips [21]. For example, a fish ticket that landed 100 pounds of fish sold at $2/pound that consisted of 10 km of fishing track would have 10 pounds of fish and $20 assigned to each km of fishing track. If that same fish ticket had 6 km of fishing tracks in block A and 4 km in block B, block A would be assigned a value of 60 pounds and $120 over 6 km for that trip, and block B would be assigned a value of 40 pounds and $80 over 4 km. We then summarized the total landings and ex-vessel value by summing the total effort, landings, and ex-vessel value for each block in each year.

We summarized data by species groupings (using the 7 species groupings described in Section 2.2) and California ports/port complexes [39]. We also evaluated Pearson correlations between the total groundfish fish ticket dataset (i.e., matched plus unmatched), the matched groundfish dataset (i.e., fish tickets that matched to VMS polls), and the unmatched groundfish dataset (i.e., fish tickets that did not match to VMS polls).

## 2.6 Visualization

To visualize our estimated metrics of VMS data and merged VMS-fish ticket (hereafter, VMS-FT) groundfish data fisheries activity (effort, landings and ex-vessel value) across all lease blocks in the region, we created heat maps using cumulative octiles, where each octile (corresponding to one color in the colorbar) represents approximately 12.5% of the sum of the fisheries activity over all the lease blocks (e.g., in a fishing effort map, lease blocks with the first and second color octiles cumulatively represent 25% of the total effort across the mapped domain). Since different fisheries have different magnitudes of a given estimated metric, using cumulative octiles allows us to display detailed spatial patterns of a given fishery in an objective and systematic way. It is important to note that hot spots of some fisheries identified by hot colors can have lower magnitude than low spots of other fisheries identified by cool colors. Lease blocks that include data from less than three unique vessels are not displayed to protect the privacy of vessel operators (e.g., the so-called "rule of three", cf. NOAA Administrative Order 216–100). Therefore, within the context of the rule of three, cell values in the first color octile do not include zero. The fisheries activity data in blocks that do not meet the rule of three are not included in the calculation of the octiles, but are included in the other, non-map figures describing our results.

## 3. Results

### 3.1 VMS

A total of 247,043 fishing trips were identified within the VMS dataset (2010–2017) across 30 valid vessel declaration codes, including four exemption declarations not shown (Table 1). Of these, 37% represented trips targeting groundfish across 16 declaration codes. Analysis of the density histograms of vessel speed for each declaration code using the local minimum method

found that the cutoff speed for fishing ranged from 3.64–6 knots (6.74–11.11 km/hr), with a median (mean) across declaration codes of 4.61 (4.74) knots (8.54 km/hr) (S2–S4 Figs). Using this cut-off speed generated 685,237 unique fishing events with a total of 6,590,718 VMS polls, which were used for calculating fishing effort.

Comparison of the spatial pattern of fishing effort derived from the local minimum speed method versus the minimum slope method found the latter to generate a lower cutoff fishing speed for all but one valid declaration code (Table 1; S2–S4 Figs). A lower cutoff always leads to a lower estimate of fishing effort, so the local minimum method yielded higher fishing effort for all but one of the declarations codes. However, the choice of method had a negligible effect on the spatial distribution of relative fishing effort: among all declaration codes, the median (mean) correlation coefficient between spatial effort estimates derived by the two methods was R = 0.99 (0.97). Therefore, all estimates of fishing effort presented and discussed below are derived using the local minimum method for identifying fishing speed.

Comparison of our estimates of groundfish fishing effort using the VMS data to estimates of effort generated independently by the NOAA Observer Program found the spatial pattern of relative fishing effort was similar (R = 0.88 for 2011–2015, R = 0.86 for 2016–2017), although the VMS data using our method had slightly higher levels of effort than those recorded in the Observer data (Fig 2). Similarly, visual inspection of Dungeness crab fisheries activity in Oregon between our dataset and data generated by Oregon Department of Fish and Wildlife (OROWindMap, https://offshorewind.westcoastoceans.org/https://offshorewind.westcoastoceans.org/) found the patterns to be congruent (Fig 3). For example, there were hotspots of high fisheries activity along the Oregon coast at Astoria and areas north and south of Coos Bay, and lower fisheries activity between Astoria and Newport. The similar spatial patterns across different datasets for the same fisheries suggests that VMS reasonably captures the spatial patterns of relative fishing activity.

We focused our mapping visualizations on the economically important groundfish fisheries by combining those declarations with similar vessel operations (fixed gear: Fig 4; trawl gear: Fig 5), along with the economically important Dungeness crab fishery (Fig 3) and maps of most of the individual declarations (S5–S25 Figs). We excluded declarations 231 and 267 because they had very low overall fishing effort and no grid cells satisfied the rule of three constraint.

Fishing effort for fixed-gear limited entry fisheries (Declaration Codes 210 and 211) was highly variable across the US West Coast, with hot spots mostly in offshore zones and within the depth limit from Central California north through Washington (Fig 4). For the groundfish trawl fisheries (Declaration Codes 230, 231, 242, and 250), effort was mostly limited to inside the depth limit north of Monterey Bay (Central California; Fig 5), with very little effort in Tribal Trawl Gear (Declaration 250; S17 Fig) and limited entry demersal trawl shorebased Individual Fishing Quota (IFQ) (Declaration 231; not mapped because no cells met the "rule of three"). For Dungeness crab (Declaration Code 261), fishing effort was concentrated along the coast and occurred across the entire US West Coast north of Point Conception (northern end of Southern California Bight), with the highest effort in Oregon and Washington (Fig 3). Some effort appeared to occur outside of the species' depth limit, although at low intensities.

Among 17 fisheries declaration codes targeting a species with a known biological depth limit, the vast majority (15) showed that >90% of the fishing effort occurred within the target species' depth limit (Table 1; also see figure maps for depth limit contours). For the remaining three declaration codes, non-groundfish trawl gear for sea cucumber (243), and open access sheephead trap or pot gear (265), which collectively represented only 1.43% of the total calculated fisheries effort, >73% of the fishing effort occurred within the target species' depth limit.

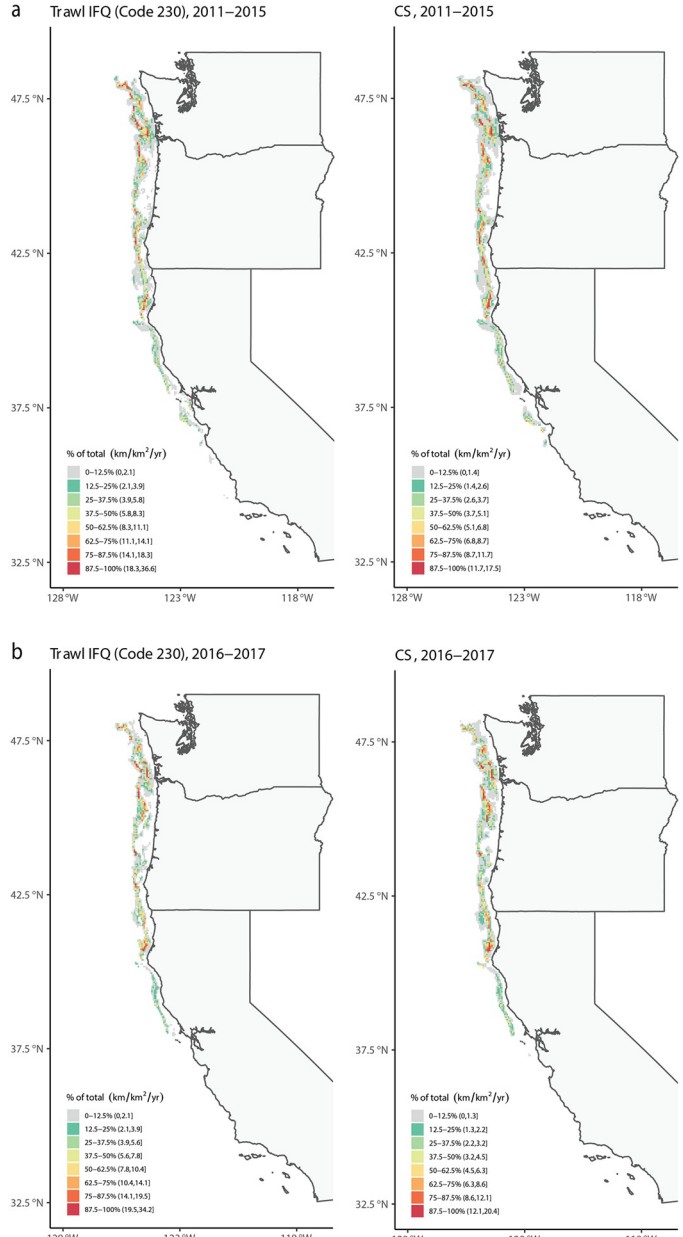

**Fig 2. Comparison of annual-mean fishing effort between VMS data and Observer data.** Left: Fishing effort from VMS data for limited entry bottom trawl shorebased individual fishing quote (IFQ) not including demersal trawl fisheries (Declaration 230). Right: Fishing effort from Observer data for catch share (CS) bottom trawl. a. 2011–2015 (R = 0.88), and b. 2016–2017 (R = 0.86). The Observer data layer is from [26], via Data Basin (https://databasin.org/datasets/8b0d742d072746ccabb98be0c9c49d8/). For comparisons between datasets with different spatial resolutions, we calculated the zonal mean of fishing efforts based on the Observer data over individual lease blocks using ArcMap 10.8.1. This step ensured a consistent spatial resolution for each dataset. Each color in the legend represents octiles of total fishing effort over the entire domain, such that the lowest octile in gray represents locations that collectively constitute the lowest 0–12.5% of the fishing effort, and the highest octile in red represents locations that collectively constitute the highest 87.5%-100% of the fishing effort. Nearly all of the maps were created using 'rnatureearth' and 'rnaturalearthdata' packages in R, which all use maps in the public domain from Natural Earth Data ((https://www.naturalearthdata.com).

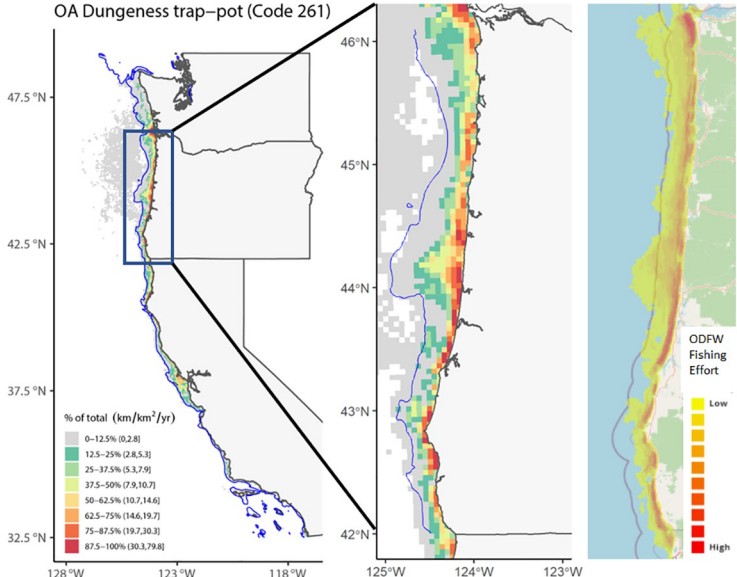

**Fig 3. Annual-mean fishing effort of Dungeness crab fishery.** Left panel: Annual-mean fishing effort (km/km²/yr) across lease blocks for open access (OA) Dungeness crab trap and pot fisheries (Declaration Code 261) during 2010–2017. The blue line is the 200 m isobath, representing the maximum biological depth for Dungeness crab. The area outside of the biological depth limit may represent erroneously reported declaration codes, and only represents 4.7% of the total effort for the fishery. Right two panels: Comparison of annual-mean fishing effort of Dungeness crab fisheries in Oregon between VMS data (zoomed-in version of the left panel) and Oregon Department of Fish and Wildlife (ODFW) Logbook data made available on OROWindMap, using an OpenStreetMap (https://www.openstreetmap.org/copyright) (rightmost).

## 3.2 Fish ticket data

Our dataset for the California groundfish fishery included 98,788 fish tickets from 2010–2017. Landings reported per ticket for a unique vessel ranged considerably, from 1 pound to 55,185 pounds; the median of all landings per ticket per vessel was 735 pounds. Individual vessels that reported a relatively small number of fish tickets over the 8 years of this dataset tended to also catch less fish per trip (S26 Fig); the vessels that reported a total of five or fewer fish tickets represented only 0.91% to the total landings, indicating that vessels that fish infrequently catch less per trip and therefore contribute very little to the total landings for the fishery. See [16] for more detailed analyses of the fish ticket dataset.

## 3.3 Matched VMS-fish ticket (VMS-FT) groundfish data

Our approach successfully matched 38,745 fish tickets with VMS trip data. Although only 39.2% of the available California groundfish fishery fish tickets were matched to VMS data, these matched fish tickets represented 87.4% and 76.3% of the total groundfish landings and ex-vessel value from the fish ticket data. Vessels fishing only in California state waters are not required to use VMS, and therefore fish tickets from those trips cannot be matched with VMS data. Other errors such as incorrect date or vessel name from fish tickets also prevented matching with VMS trips. In the total fish ticket data, 96.6% of the vessels (1,990 out of 2,061) reported relatively small or moderate landings (<100 metric tons); after matching this percentage dropped to 89% (511 out of 574 vessels). Consequently, the matched data contains relatively more vessels with a large number of fish ticket records and landings than the before-match data. Approximately 15% of VMS polls in the matched data were from non-groundfish

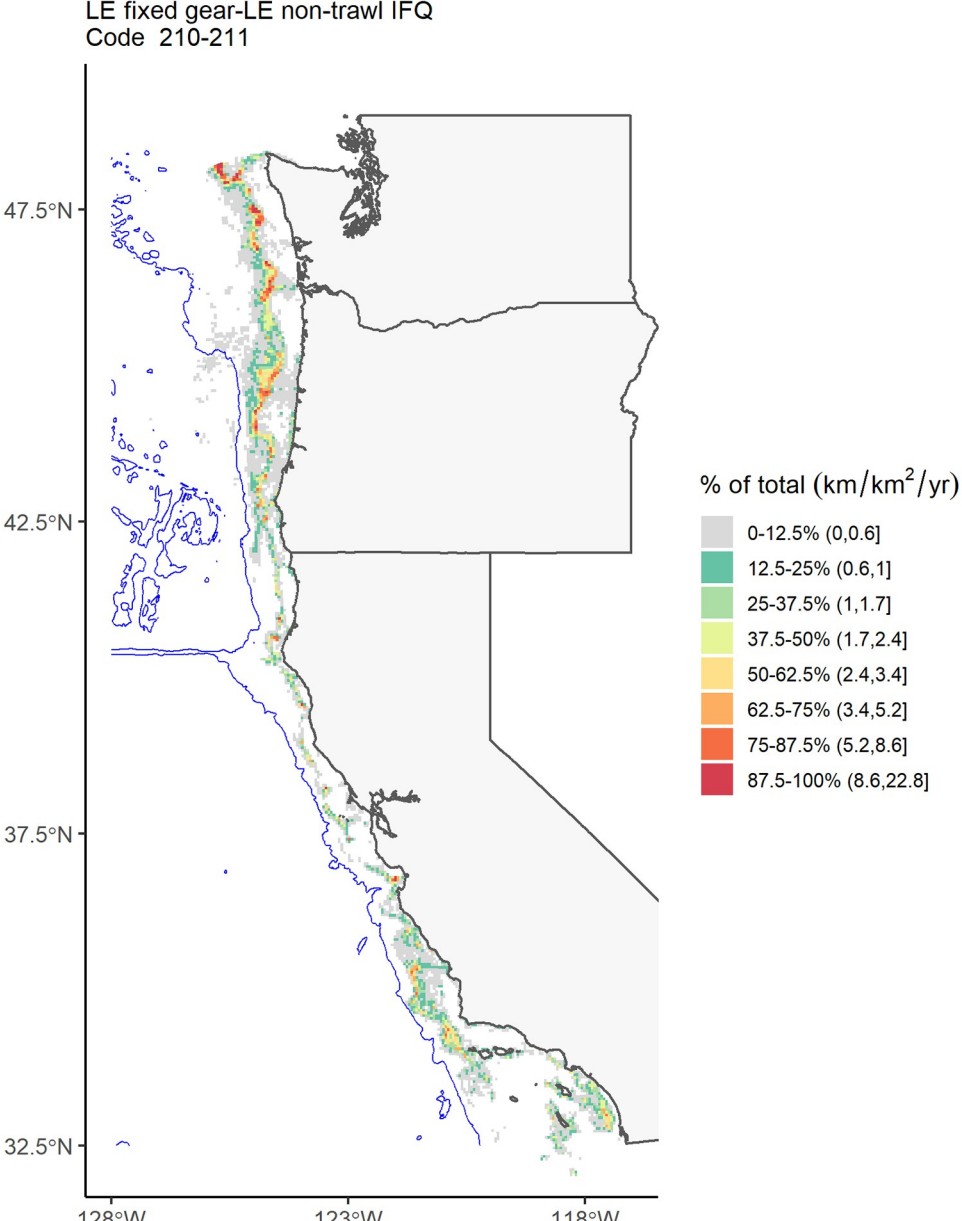

**Fig 4. Annual-mean fishing effort across lease blocks for fixed-gear limited entry (LE) individual fishing quota (IFQ) fisheries (Declaration Codes 210 and 211).** The unit is km/km$^2$/yr. The blue line is the 2700 m isobath, representing the maximum depth for all groundfish fisheries except for midwater trawl. The area outside of the biological depth limit may represent erroneous declaration codes, and only represent 2.3% of the total effort for the fishery.

fishery declaration codes, with the most (4.7% of VMS polls) attributed to the Dungeness crab fishery (Declaration Code 261), but since declaration codes were not considered in the matching process, these presumably erroneous declaration codes had no impact on the matched dataset.

In addition, data aggregated either at the groundfish species group level or port level showed reasonably high correlations for landings and ex-vessel value when compared among

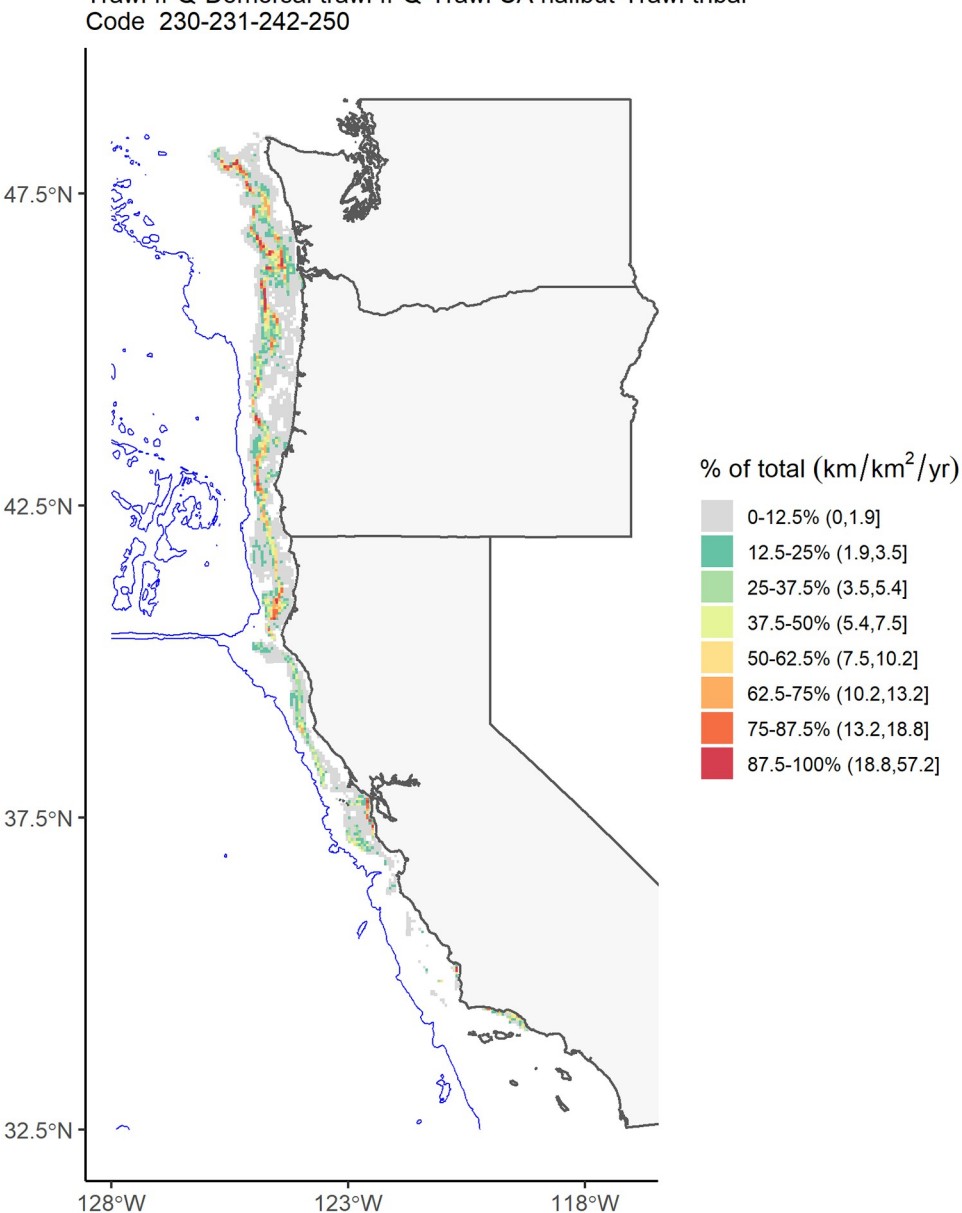

**Fig 5. Annual-mean fishing effort for groundfish trawl fisheries, including individual fishing quota (IFQ), demersal trawl IFQ, California halibut trawl, and tribal trawl fisheries (Declaration Codes 230, 231, 242, and 250).** Similar to Fig 4. Almost all of the data (~100%) are inside the biological depth limit for the species.

the total groundfish fish ticket dataset, the matched data, and unmatched data (Table 2). Note that there are 7 species groups for all comparisons, but the number of ports varied depending on the subset used (n = 85 ports for the total dataset, n = 48 ports for the matched data, and n = 81 ports for the unmatched data). To compare the species and port characteristics, we performed correlation analysis. For a port without any reported landings or ex-vessel value, the landings and ex-vessel value are set to zero for that port. The correlations between the full groundfish dataset and the matched data were very high (R >= 0.98 in all cases; Table 2). However, the correlations between the total dataset and matched data with the unmatched data

**Table 2. Correlation coefficient of species and port composition of groundfish fisheries.**

| R | Species: landing (n = 7) | Species: ex-vessel value (n = 7) | Port: landing (n = 85) | Port: ex-vessel value (n = 85) |
|---|---|---|---|---|
| All vs matched | 0.997 | 0.98 | 0.998 | 0.99 |
| All vs unmatched | 0.57 | 0.55 | 0.65 | 0.71 |
| Matched vs unmatched | 0.51 | 0.38 | 0.60 | 0.58 |

This table presents the correlation coefficient (R) between two datasets from total groundfish fish tickets (all), fish tickets after matching with VMS data (matched), and fish tickets that had no match with VMS data (unmatched). Data points for each comparison were for species groups (n = 7) or ports (n = 85). For example, for 'Species: Landing all vs matched', the R was computed using the landing weights from all data and matched data for each of the seven species groups. For ports without any reported landings or ex-vessel value, the landings and ex-vessel value were set to zero.

were lower (R values between 0.38 to 0.60). The rank order of functional groups by biomass landing and ex-vessel value was only partially in alignment between the matched and unmatched fish ticket data, but identical between the matched and total fish ticket data (Fig 6). Likewise, highest landings from unmatched fish ticket data were from Central California (Morro Bay, Avila/Port St. Luis), while the matched and total fish ticket data both showed the highest landings from two Northern California ports (Eureka, Fort Bragg) (Fig 7).

Groundfish fisheries effort was distributed unevenly latitudinally along the California coast (Fig 8), with a large hotspot in Northern California offshore of Humboldt County, and smaller hotspots along the Santa Barbara and Ventura County coasts, including the western Channel Islands. Landings and ex-vessel value spatial distributions showed similar trends to fisheries effort (Figs 9 and 10). These broad patterns generally matched those from previous work that examined spatial distribution of landings using only self-reported fishing block data from fish tickets (Wang et al. 2022). However, a more detailed analysis of individual trips from the matched dataset for groundfish found that 64% of the trips had no VMS polls recorded from the self-reported fishing block recorded in the fish ticket.

To gain a deeper insight into the temporal and spatial dynamics of fishing activities, we presented fishing effort, landings, and ex-vessel value for each year across depth ranges in S27–S29 Figs. To create these figures, we first computed the mean ocean depth within each lease block. Then, for each year of matched groundfish fisheries data, we summed each of fishing effort, landing, and ex-vessel value over the lease blocks within the respective depth intervals. Averaged across the state of California, fishing effort by the matched groundfish fisheries data was found to be greatest closest to shore (< 100 m depth), with a second, much smaller maximum offshore at ~500–600 m depth (S27 Fig). Fisheries landings and ex-vessel value also exhibited local maxima close to shore at < 100 m depth and at ~500–600 m depth (S28 and S29 Figs). Fisheries effort, landings, and ex-vessel value all exhibited local minimums between these depths around ~200–300 m (S28 and S29 Figs).

To understand interannual variability of fishing activities, the matched data were categorized on a yearly basis. State-wide groundfish fisheries activity varied substantially over the eight-year evaluation period, with an annual change as large as 30% (from 2011 to 2012) (S30 Fig). Fishing effort, landings and ex-vessel value all declined precipitously from peak levels early in the time series (2010–11), followed by a further but much weaker decline in effort over the remainder of the time series, coincident with moderately variable but loosely stable landings and positive-trending ex-vessel value.

## 4. Discussion and conclusions

We used a combination of federally-managed spatial fisheries data (VMS) and state-managed landings data (fish tickets) to quantify spatial patterns in fisheries effort, catch, and ex-vessel

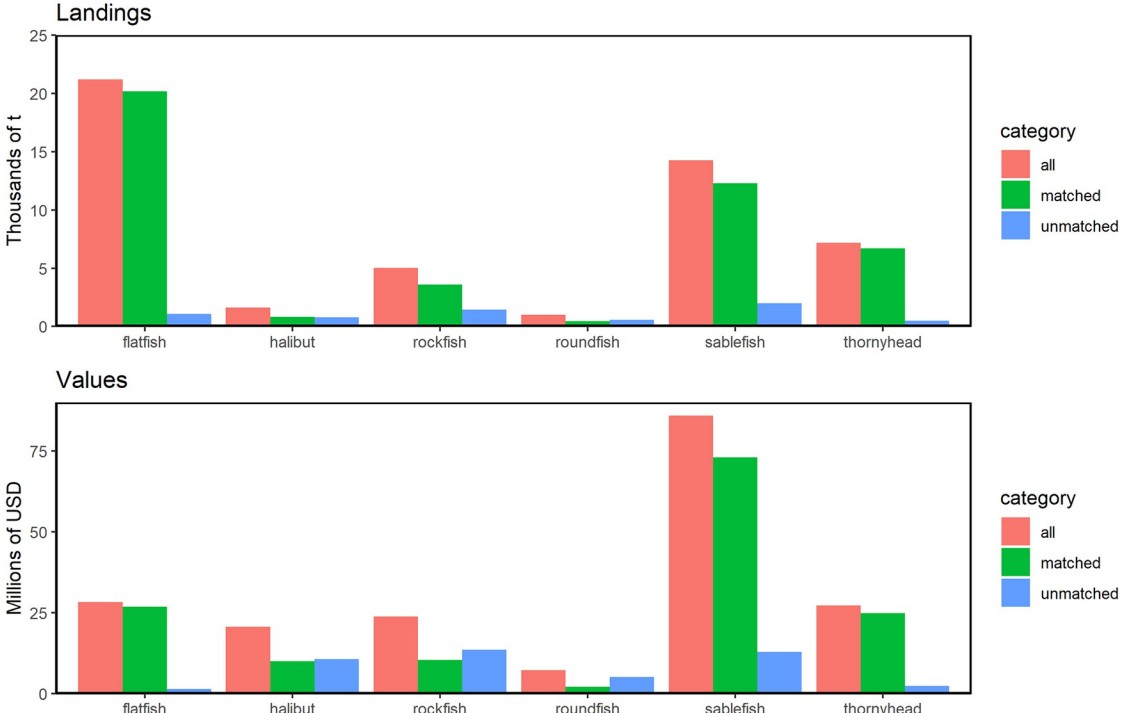

**Fig 6. Landings and ex-vessel values of groundfish fisheries categorized by species.** Landings (top, in metric tons, t) and ex-vessel values (bottom, in 2019 dollars) of all groundfish fisheries in California from 2010–2017, categorized into 7 species subsets, based on total fish tickets ('all'), fish tickets that matched with VMS trips ('matched'), and fish tickets that did not match with VMS trips ('unmatched'). Species groups are ordered alphabetically. Note that 'other' is not shown because landings and ex-vessel values were too low to appear on the plot. For each group, 'all' is the sum of 'matched' and 'unmatched'.

value for the economically-important groundfish fisheries in California, as well as spatial patterns in effort across a range of other fisheries throughout the US West Coast. As expected, the data revealed substantial spatial variability in fishing effort within and among different fisheries. Importantly, our estimates of effort generated using VMS data for certain groundfish fisheries matched closely with estimates for the same fisheries using NOAA Observer data. Since the NOAA Observer dataset has nearly complete coverage, this provides evidence that our estimates of effort derived from VMS data accurately capture the spatial pattern of relative fishing activities. This is particularly true for US West Coast groundfish fisheries, given that VMS is legally required and Observer datasets for groundfish fisheries have greater than 99% coverage [38]. Additionally, the spatial patterns from VMS data for the Dungeness crab fishery, for which VMS is only required if they hold a permit or participate in groundfish fishing in that year (e.g., approximately 19–26% of the Dungeness crab fishery's vessels and 10–57% of its landings; [29]), closely match the independently-generated estimates of effort from ODFW This suggests that VMS data may be useful for evaluating US West Coast fisheries without high-coverage VMS data. This validation lends important credibility to the high-resolution estimates of catch and ex-vessel value generated here. As such, we expect these data to be highly useful in supporting fisheries management and marine spatial planning.

### 4.1 US West Coast fisheries effort based on VMS data only

Since the VMS data primarily generates information on location only, vessel speed must be inferred from distance traveled between two points. Therefore, one of the most critical

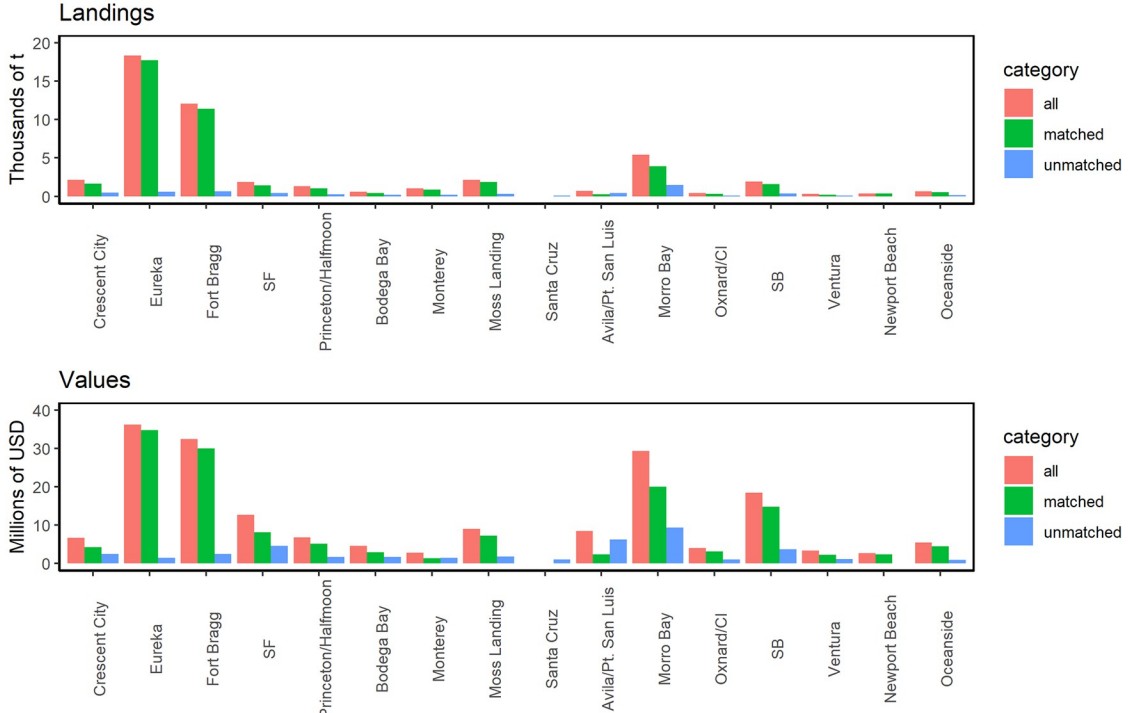

**Fig 7. Landings and ex-vessel values of groundfish fisheries categorized by ports.** Similar to Fig 6, but for the top 15 ports in California.

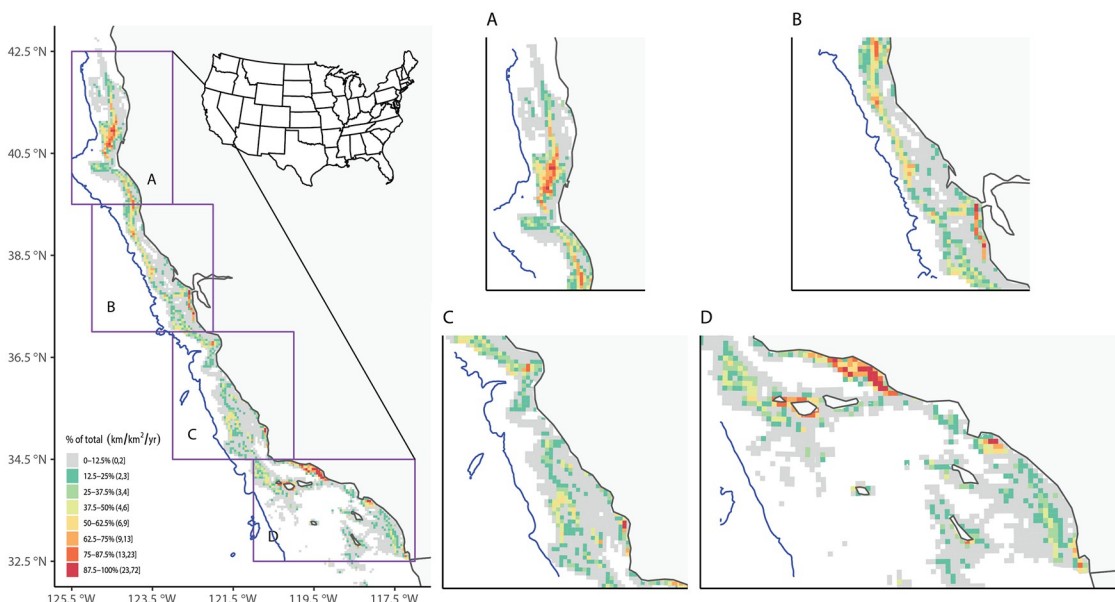

**Fig 8. Annual-average fishing effort using matched VMS-landings data for groundfish.** Left: Annual-average fishing effort (in km/km$^2$/yr) in BOEM lease blocks along the California coast over 2010–2017. The blue line is the 2700 m isobath, representing the maximum depth for all groundfish fisheries. Right: Same as left, but with regions magnified.

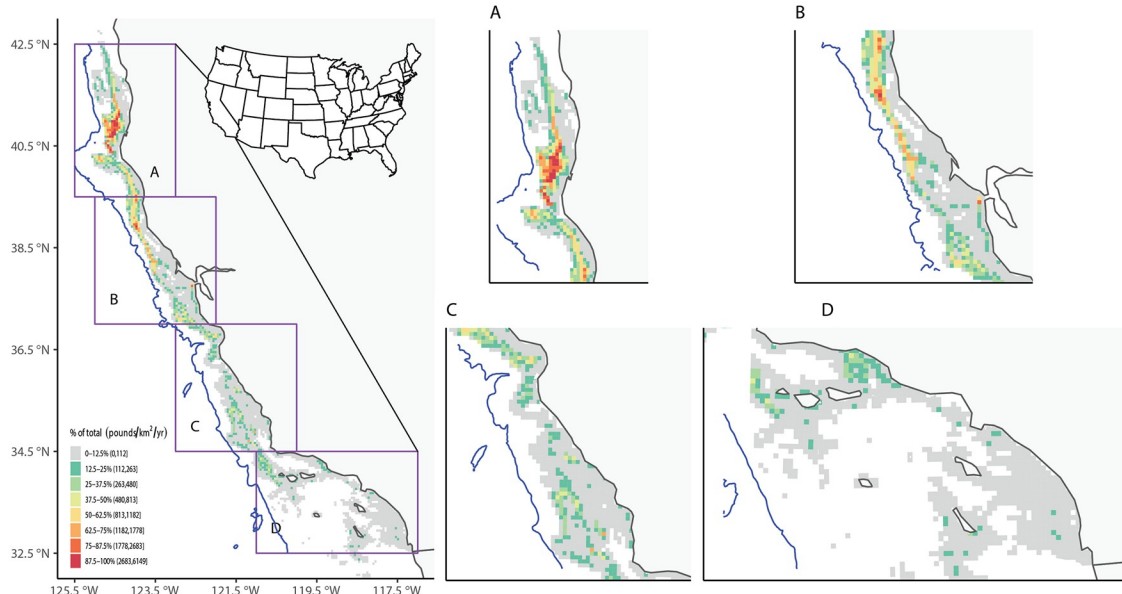

**Fig 9. Annual-average landings using matched VMS-landings data for groundfish.** Similar to Fig 8, except for annual-average landings (in lbs/km²/yr).

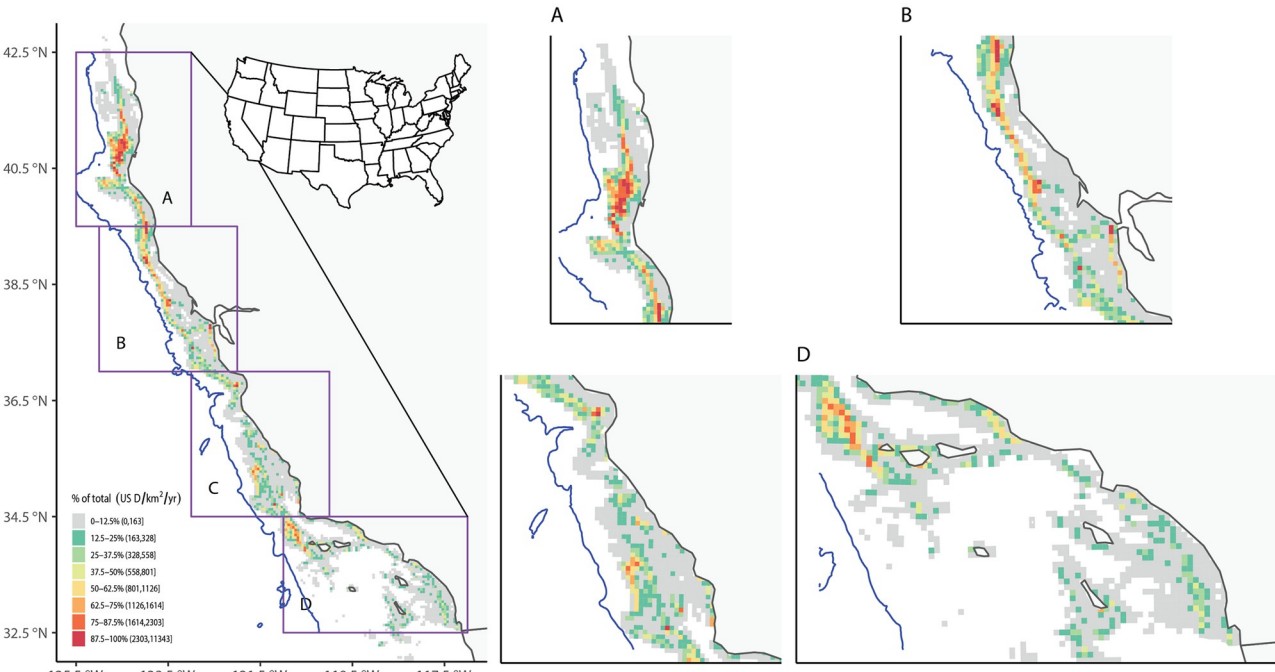

**Fig 10. Annual-average ex-vessel value using matched VMS-landings data for groundfish.** Similar to Fig 8, except for ex-vessel value (in 2019 dollars/km²/yr).

challenges is determining vessel activity (i.e., fishing or transit) from these data. Despite these challenges (see e.g., [40]), our estimates of the spatial distribution of relative fishing effort were largely insensitive to the assumed cutoff speed between fishing and transit, based on two methods of estimating cutoff speed, suggesting that our results are robust to underlying assumptions about vessel activity. Furthermore, our analyses of VMS data found that a much higher percentage of fishing effort occurred within the biological depth limits for a given species than the self-reported block data from fish ticket data alone [16]. These findings, combined with the strong correlation between spatial patterns between VMS data and NOAA Observer data, highlight the reliability of our VMS-based estimates of spatial fisheries effort.

Estimates of fisheries effort based on fish ticket data alone found small but significant inaccuracies revealed by fish ticket reports of landings beyond the species' biological depth limit (Wang et al. 2022). Furthermore, we also examined the correspondence between reported fishing blocks from the fish ticket dataset and the track recorded from VMS data. This analysis found that 64% of the groundfish fishing trips from the matched dataset did not have a single VMS poll inside the fishing block reported in the fish ticket, suggesting that that self-reported fishing block data are highly unreliable. Combined with the reliability of VMS data in recording vessel location, these findings suggest that VMS data can more accurately map fisheries activities. In addition, this highlights the need to improve the reliability of self-reported fishing location information in fish tickets for that data stream to be useful in understanding the spatial distribution of fisheries. Our analyses revealed a number of noteworthy spatial patterns in effort across a range of fisheries (see S5–S25 Figs for fishing effort maps across all VMS declarations). Fishing effort is strongly heterogeneous along the US West Coast, typically with greater effort in northern regions (Oregon and Washington) than further south (California). This pattern is especially clear for midwater trawl fisheries (S7 and S8 Figs) and HMS fisheries (S23 Fig), both of which are concentrated in Oregon and Washington, with little activity in California. Several fisheries also clearly target specific depth bands, either nearshore (e.g., open access Dungeness crab, Fig 3) or offshore (e.g., trawl gear for pink shrimp, S14 Fig; open access line gear for HMS, S23 Fig; and fixed gear and trawl fisheries for groundfish, Figs 4 and 5). Other species appear to be highly concentrated either latitudinally or in relation to depth, or in very specific local areas. For example, ridgeback prawn and sea cucumber trawl fisheries are limited to the nearshore in the Southern California Bight (S13 and S16 Figs), and open access trap fisheries for spot prawn are limited to offshore areas in northern Oregon (S18 Fig). It is important to note that these non-groundfish fisheries do not require VMS, so estimates from VMS data alone likely do not capture the absolute magnitude of fishing effort. While the spatial patterns for selected bottom trawl and Dungeness crab fisheries in the VMS dataset generally matched those from other datasets, the spatial patterns revealed by the VMS dataset for each of these other fisheries may or may not be broadly representative of activity for that fishery.

## 4.2 California groundfish fisheries patterns using merged VMS-FT dataset

Based on fish tickets recording landings of groundfish, large vessels tend to generate more contribution to the overall landings (S26 Fig). The matched VMS-FT dataset (see Method and Results details in sections 2.4 and 3.3) included a higher proportion of vessels with high total weight landings than the full fish ticket dataset, and therefore included fewer vessels that fished less and landed smaller quantities of fish in each trip. It is possible that many of the smaller vessels with fewer fish tickets and/or less total fish landed were fishing closer to shore, possibly only in state waters (<3 nm [~5.6 km] from shore), such that VMS was not required. Since fish tickets from many of those vessels could not be matched to VMS trips, their activity was

not included in the matched VMS-FT dataset. Nonetheless, the greatest amount of fishing effort recorded in the matched VMS-FT data occurred in depths of less than 100 m (S27 Fig). In most locations in California, 100 m is well within state waters, indicating that a substantial proportion of the overall groundfish fishing activity is occurring in state waters by vessels with VMS. Consequently, the absence of data from small-vessel, state water-only fishing activity may not substantially impact the overall estimates of spatial and temporal patterns of the groundfish fishery. The absence of this information also should have virtually no impact on estimates of fishing activity occurring in federal waters.

Importantly, the matched dataset is very similar in a variety of attributes to the full fish ticket dataset: the relative contributions of different species and different ports for both landings and ex-vessel value are very similar in both datasets, and the matched dataset captures a large proportion of the landings and ex-vessel value in the full dataset (87% and 76%, respectively), even though the matched dataset includes only 39% of the groundfish fish tickets (Figs 6, 7; Table 2). Overall, these findings suggest that the matched fish ticket dataset provides an excellent representation of the commercial groundfish fishery activity in federal waters of California. While the matched data also shows significant activity in state waters, it will be important to interpret these inshore patterns carefully if they are used for spatial management or planning purposes, since this combined dataset may be missing different components of groundfish fishing activity by vessels operating only in state waters.

For groundfish fisheries, both effort and ex-vessel value showed increasing intensity close to shore and again offshore at ~500–600 m depth, while landings peaked offshore at ~500–600 m depth without a second clear nearshore local maximum (S27–S29 Figs). Despite some differences, effort, landing, and ex-vessel value all showed a consistent presence of a local minimum at ~200–300 m depth occurring across the state and over the eight-year study period (S27–S29 Figs), which may be partly attributable to the boundaries of the Rockfish Conservation Areas (RCA) and other areas that limit groundfish trawling in that depth zone [41, 42]. The RCAs in California have changed over time in response to changes in the status of stocks they are meant to protect. As certain stocks continue to recover, these closed areas for trawling have been reduced or removed over the last several years [43], providing the opportunity to evaluate changes in fishing activity that may expand into the previously-closed 200-300m depth zone (S27–S29 Figs).

The groundfish fisheries in California target a range of different species with differing life histories, habitat requirements, and population dynamics in different regions of the state (e.g., [44]). As a result, fisheries across the state target multiple high-value species. Fisheries activities fluctuated over time (S30 Fig), including a noticeable drop in 2012 that may be linked to the implementation of trawl rationalization (i.e., IFQ program; [45]). But the temporal changes in this eight-year study period were relatively small compared to changes observed over larger time scales (e.g., [15, 16]).

## 4.3 Caveats and future research

As we demonstrate, VMS-based estimates of fisheries metrics are likely highly reliable, but there are still important limitations to these data. First, not all vessels are required to utilize VMS, so the actual absolute magnitude of a given metric (e.g., total effort, at any scale) may be underestimated for many declaration codes, with the possible exception of the commercial groundfish fisheries, for which VMS is required (see section 2.1). However, while it may be difficult to quantify the absolute magnitude of fishing intensity at a given location, the relative spatial patterns are likely representative, such that areas of high and low fishing intensity are generally accurately identified. Second, the fisheries effort metric of km/km$^2$ used here is likely

more amenable for measuring mobile gear fisheries (such as trawls, that travel while fishing), and less ideal for measuring fixed gear fisheries (e.g., pots). Therefore, further validation is needed to allow direct comparisons of relative patterns of effort between fisheries. Third, we focused on commercial fisheries activity predominantly occurring in federal waters monitored with VMS, while recognizing that other fisheries outside the scope of this study contribute significant landings and substantial socioeconomic value to the US West Coast. For example, recreational fisheries constitute a smaller but significant proportion of the total fisheries harvest in California, Oregon and Washington, with disproportionate overall economic impact [46], and subsistence fisheries support a diversity of communities and local economies across the US West Coast [47]. In addition, there are some fisheries that operate nearly-exclusively in state waters, such as the nearshore live-fish fisheries [48]; because they are not operating in federal waters, they are not monitored by VMS. While the number of operators engaged in that fishery and landings have declined in recent years [49], the data we present here will not capture any of that activity. An improved understanding of these fisheries alongside the results presented here for most California commercial fisheries is needed to develop a more holistic understanding of the spatial patterns and socioeconomic contributions to communities throughout the West Coast region. Fourth, while most of the activity for a given fishery occurred within the biological limits of the target species, some activity for some fisheries occurred outside these limits (Table 1). Such activity is likely the result of misidentified declaration codes (e.g., a vessel trip declares a Dungeness crab code but is actually fishing offshore for HMS). While NOAA OLE attempts to correct these obvious errors in real-time, they are unable to fix all incorrect declaration codes (K. Spalding, pers. comm.). Despite these issues, in nearly all cases the likely erroneous declaration codes outside of the depth limits comprise less than 5% of the total effort for the fishery, such that the broad spatial patterns these data represent are likely accurate.

Integrating additional spatial fisheries datasets (e.g., NOAA Observer data, state-level Logbook data) with the data generated here could help validate and further calibrate our findings. Moreover, the matched VMS-FT groundfish dataset only included landings data from California. Future research could expand to include Oregon and Washington, which have similar landings receipt data systems, as well as incorporate NMFS Observer and Logbook datasets. These coast-wide datasets could be used to further assess spatial patterns, landings, and ex-vessel value across all federally-managed US West Coast groundfish fisheries, all of which require VMS coverage. Such an expansion could support efforts to inform a range of important spatial and fishery management questions.

In this study, we developed methods to generate the combined VMS-FT data that, to our knowledge, is one of the highest spatial resolution and most accurate assessments of recent fishery activity along the entire US West Coast (but see [20, 27, 28] for recent applications of these approaches on select species). There are numerous potential future applications for these data in marine science and management, including spatially-explicit fishery stock assessments and ecosystem-based management, marine spatial planning of protected areas and new uses of ocean space such as offshore aquaculture and renewable energy [50], assessing spatial human-wildlife conflicts [20, 28], and supporting climate-change readiness and resilience planning [51].

## 4.4 Conclusion

Fine-scale spatial characterization of fishery dynamics is increasingly important for supporting fisheries management, from regional stock assessment to ecosystem-based approaches to spatial regulation and planning [52]. In addition, other uses of ocean space have intensified over the

last few decades, including increased shipping activity, offshore aquaculture, and offshore renewable energy development. These uses may interact with fisheries in significant, new ways [53, 54]. Concurrently, advances in fisheries remote monitoring technology are generating a wealth of information on the movement, distribution, activities, and impacts of fishing vessels at unprecedented spatio-temporal scales (e.g., [55]). This study demonstrates a process to obtain, process, integrate and analyze such data—fish ticket reports and VMS polls—to provide a more comprehensive and up-to-date understanding of commercial fisheries activities across the US West Coast. Our hope is that the products generated here will provide valuable support for spatial fisheries assessment and management, including efforts in ecosystem-based fisheries science to sustainably integrate fisheries with new ocean uses along the US West Coast.

## Supporting information

**S1 Fig. Sankey diagram representing VMS data loss in each step of processing.** The starting point in the diagram is the raw data after correcting negative latitudes (0.39% of raw data). The percentage of data relative to the raw data for each step is provided. The raw data included over 50 million VMS polls, so the data for downstream analysis of fishing activity (12.5% of the VMS polls) still included over 6 million individual VMS polls.
(TIF)

**S2 Fig. Distribution of vessel speed for Declaration Codes 210–233 using processed VMS data.** The blue line represents a kernel density fit of the data. The red solid line shows the local minimum of the distribution and the red dashed line shows the slope minimum between two and six knots. The number in the title for each subplot is the declaration code, and no. pts is the number of VMS polls included in each declaration code.
(TIF)

**S3 Fig. Distribution of vessel speed for Declaration Codes 234–261 using processed VMS data.** Similar to S2 Fig, but for Declaration Codes 234–261.
(TIF)

**S4 Fig. Distribution of vessel speed for Declaration Codes 262–269 using processed VMS data.** Similar to S2 Fig, but for Declaration Codes 262–269.
(TIF)

**S5 Fig. Annual-mean fishing effort per unit area (km fished/km$^2$/yr) across lease blocks for fixed limited entry (LE) fisheries not including shorebased IFQ (Declaration Code 210).** The blue line is the 2700 m isobath, representing the maximum depth for all groundfish fisheries except for midwater trawl. The area outside of the biological depth limit may represent erroneous declaration codes, and only represent 2.6% of the total effort for the fishery.
(TIF)

**S6 Fig. Annual-mean fishing effort per unit area (km fished/km$^2$/yr) across lease blocks for limited entry groundfish non-trawl shorebased IFQ (Declaration Code 211).** The area outside of the biological depth limit may represent erroneous declaration codes, and only represent 0.8% of the total effort for the fishery.
(TIF)

**S7 Fig. Annual-mean fishing effort per unit area (km fished/km$^2$/yr) across lease blocks for limited entry midwater trawl (MW) gear non-whiting shorebased IFQ (Declaration Code 220).** An isobath line is not added since it is not applicable to the midwater trawl fishery.
(TIF)

**S8 Fig. Annual-mean fishing effort per unit area (km fished/km$^2$/yr) across lease blocks for mid-water trawl Pacific whiting, including whiting IFQ, whiting catcher-processor, and whiting mothership fisheries (Declaration Codes 221, 222, and 223).** An isobath line is not including since depth limits are not applicable to the midwater trawl fisheries.
(TIF)

**S9 Fig. Annual-mean fishing effort per unit area (km fished/km$^2$/yr) across lease blocks for limited entry bottom trawl shorebased IFQ not including demersal trawl (Declaration Code 230).** Almost all of the data (~100%) are inside the biological depth limit for the species.
(TIF)

**S10 Fig. Annual-mean fishing effort per unit area (km fished/km$^2$/yr) across lease blocks for open access (OA) longline gear for groundfish (Declaration Code 233).** The area outside of the biological depth limit may represent erroneous declaration codes, and only represent 2.7% of the total effort for the fishery.
(TIF)

**S11 Fig. Annual-mean fishing effort per unit area (km fished/km$^2$/yr) across lease blocks for open access (OA) groundfish trap or pot gear (Declaration Code 234).** The area outside of the biological depth limit may represent erroneous declaration codes, and only represent 1.1% of the total effort for the fishery.
(TIF)

**S12 Fig. Annual-mean fishing effort per unit area (km fished/km$^2$/yr) across lease blocks for open access (OA) line gear for groundfish (Declaration Code 235).** The area outside of the biological depth limit may represent erroneous declaration codes, and only represent 6% of the total effort for the fishery.
(TIF)

**S13 Fig. Annual-mean fishing effort per unit area (km fished/km$^2$/yr) across lease blocks for non-groundfish trawl gear for ridgeback prawn (Declaration Code 240).** The blue line is the 300 m isobath, representing the maximum biological depth for the target species. The area outside of the biological depth limit may represent erroneous declaration codes, and only represent 0.4% of the total effort for the fishery. Data were too sparse to generate octiles as in the other plots.
(TIF)

**S14 Fig. Annual-mean fishing effort per unit area (km fished/km$^2$/yr) across lease blocks for non-groundfish trawl gear for pink shrimp (Declaration Code 241).** The blue line is the 300 m isobath, representing the maximum biological depth for the target species. The area outside of the biological depth limit may represent erroneous declaration codes, and only represent 1.7% of the total effort for the fishery.
(TIF)

**S15 Fig. Annual-mean fishing effort per unit area (km fished/km$^2$/yr) across lease blocks for non-groundfish trawl gear for California halibut (Declaration Code 242).** The blue line is the 200 m isobath, representing the maximum biological depth for the target species. The area outside of the biological depth limit may represent erroneous declaration codes, and only represent 0.4% of the total effort for the fishery.
(TIF)

**S16 Fig. Annual-mean fishing effort per unit area (km fished/km$^2$/yr) across lease blocks for non-groundfish trawl gear for sea cucumber (Declaration Code 243).** The blue line is

the 100 m isobath, representing the maximum biological depth for the target species. The area outside of the biological depth limit may represent erroneous declaration codes, and represent 10.3% of the total effort for the fishery.
(TIF)

**S17 Fig. Annual-mean fishing effort per unit area (km fished/km$^2$/yr) across lease blocks for tribal trawl gear (Declaration Code 250).** All of the data (100%) are inside the biological depth limit for the species. Data were too sparse to generate octiles as in the other plots.
(TIF)

**S18 Fig. Annual-mean fishing effort per unit area (km fished/km$^2$/yr) across lease blocks for open access prawn trap or pot gear (Declaration Code 260).** The blue line is the 300 m isobath, representing the maximum biological depth for the target species. The area outside of the biological depth limit may represent erroneous declaration codes, and only represent 3.2% of the total effort for the fishery.
(TIF)

**S19 Fig. Annual-mean fishing effort per unit area (km fished/km$^2$/yr) across lease blocks for open access Pacific Halibut longline gear (Declaration Code 262).** The area outside of the biological depth limit may represent erroneous declaration codes and represent 4.9% of the total effort for the fishery.
(TIF)

**S20 Fig. Annual-mean fishing effort per unit area (km fished/km$^2$/yr) across lease blocks for open access salmon troll gear (Declaration Code 263).** An isobath line is not added since it is not applicable to the salmon troll fishery.
(TIF)

**S21 Fig. Annual-mean fishing effort per unit area (km fished/km$^2$/yr) across lease blocks for open access California halibut line gear (Declaration Code 264).** The blue line is the 200 m isobath, representing the maximum biological depth for the target species. The area outside of the biological depth limit may represent erroneous declaration codes, and only represent 6.3% of the total effort for the fishery.
(TIF)

**S22 Fig. Annual-mean fishing effort per unit area (km fished/km$^2$/yr) across lease blocks open access for sheephead trap or pot gear (Declaration Code 265).** The blue line is the 60 m isobath, representing the maximum biological depth for the target species. The area outside of the biological depth limit may represent erroneous declaration codes, and represent 26.2% of the total effort for the fishery. Data were too sparse to generate octiles as in the other plots.
(TIF)

**S23 Fig. Annual-mean fishing effort per unit area (km fished/km$^2$/yr) across lease blocks for open access Highly Migratory Species (HMS) line gear (Declaration Code 266).** An isobath line is not added since it is not applicable to the HMS fishery.
(TIF)

**S24 Fig. Annual-mean fishing effort per unit area (km fished/km$^2$/yr) across lease blocks for open access California gillnet complex gear (Declaration Code 268).** An isobath line is not added since it is not applicable to the gillnet fishery.
(TIF)

**S25 Fig. Annual-mean fishing effort per unit area (km fished/km$^2$/yr) across lease blocks for gear not listed (Declaration Code 269).** An isobath line is not added since it is unknown what species are captured with unlisted gear.
(TIF)

**S26 Fig. Frequency distribution of groundfish landings per fish ticket record for vessels with different numbers of fish ticket submission during 2010–2017.** Panels from top to bottom represent vessels reporting fish total ticket records between 1 and 5, between 6 and 10, between 11 and 50, and greater than 50. (a) Total fish ticket records. (b) Fish ticket records that matched with VMS (matched). The unit of landings is pounds. Note that the vessels that reported 5 or fewer fish tickets also reported lower landings per fish ticket than those that reported >50 fish tickets. That is, vessels that reported a relatively small number of fish tickets tended to also catch less fish per trip.
(TIF)

**S27 Fig. Fishing effort by year and depth ranges scaled by the maximum of the year based on VMS-FT data for groundfish.** The top panel shows the average of the scaled fishing effort in the bottom panel over years.
(TIF)

**S28 Fig. Landings by year and depth ranges scaled by the maximum of the year based on VMS-FT data for groundfish.**
(TIF)

**S29 Fig. Ex-vessel value by year and depth ranges scaled by the maximum of the year based on VMS-FT data for groundfish.**
(TIF)

**S30 Fig. Annual fishing effort, landings, and ex-vessel value relative to respective maximum.** The maximum for effort (red) is 187.89 million km for effort in 2010. Based on VMS-FT data for groundfish, the maximum for landings (green) is 14.66 million pounds for landing in 2010. The maximum for ex-vessel value (blue) is $23.23 million in 2011.
(TIF)

## Acknowledgments

The views and conclusions contained in this document are those of the authors and should not be interpreted as representing the opinions or policies of the U.S. Government. Mention of trade names or commercial products does not constitute their endorsement by the U.S. Government. Study collaboration was provided by the U.S. Department of the Interior, Bureau of Ocean Energy Management. We are indebted to Brian Owens, Todd Neahr and Paulo Serpa at the California Department of Fish and Wildlife (CDFW) for providing access to and sharing insights about CDFW landings data. CDFW acquires data from its own fisheries management activities and from mandatory reporting requirements on the commercial and recreational fishery pursuant to the Fish and Game Code and the California Code of Regulations. These data are constantly being updated, and data sets are constantly modified. CDFW may provide data upon request, but, unless otherwise stated, does not endorse any particular analytical methods, interpretations, or conclusions based upon the data it provides. We also thank Kelly Spaulding in the National Marine Fisheries Service Office of Law Enforcement for providing access to Vessel Monitoring System Data.

## Author Contributions

**Conceptualization:** Yi-Hui Wang, Benjamin I. Ruttenberg, Ryan K. Walter, Crow White.

**Data curation:** Yi-Hui Wang, Benjamin I. Ruttenberg, Ryan K. Walter, Frank Pendleton, Jameal F. Samhouri, Owen R. Liu, Crow White.

**Formal analysis:** Yi-Hui Wang.

**Funding acquisition:** Benjamin I. Ruttenberg.

**Investigation:** Yi-Hui Wang, Benjamin I. Ruttenberg, Ryan K. Walter, Crow White.

**Methodology:** Yi-Hui Wang, Benjamin I. Ruttenberg, Ryan K. Walter, Jameal F. Samhouri, Owen R. Liu, Crow White.

**Project administration:** Benjamin I. Ruttenberg.

**Resources:** Benjamin I. Ruttenberg.

**Software:** Benjamin I. Ruttenberg.

**Supervision:** Benjamin I. Ruttenberg.

**Validation:** Yi-Hui Wang.

**Visualization:** Yi-Hui Wang, Frank Pendleton.

**Writing – original draft:** Yi-Hui Wang.

**Writing – review & editing:** Benjamin I. Ruttenberg, Ryan K. Walter, Frank Pendleton, Jameal F. Samhouri, Owen R. Liu, Crow White.

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
