## [Decision Letter · Decision Letter 0]

18 Sep 2023

PONE-D-23-21225High resolution assessment of commercial fisheries activity along the US West Coast using Vessel Monitoring System dataPLOS ONE

Dear Dr. Ruttenberg,

Thank you for submitting your manuscript to PLOS ONE. After careful consideration, we feel that it has merit but does not fully meet PLOS ONE’s publication criteria as it currently stands. Therefore, we invite you to submit a revised version of the manuscript that addresses the points raised during the review process.

We look forward to receiving your revised manuscript.

Kind regards,

Md. Naimur Rahman

Academic Editor

PLOS ONE

“The views and conclusions contained in this document are those of the authors and should not be interpreted as representing the opinions or policies of the U.S. Government. Mention of trade names or commercial products does not constitute their endorsement by the U.S. Government. Study collaboration and funding were provided by the U.S. Department of the Interior, Bureau of Ocean Energy Management, Environmental Studies Program, Washington, DC under Agreement Number Ml6AC0002. Additional funding was provided the California Ocean Protection Council (OPC), Agreement Number C0210403. We are indebted to Brian Owens, Todd Neahr and Paulo Serpa at the California Department of Fish and Wildlife (CDFW) for providing access to and sharing insights about CDFW landings data. CDFW acquires data from its own fisheries management activities and from mandatory reporting requirements on the commercial and recreational fishery pursuant to the Fish and Game Code and the California Code of Regulations. These data are constantly being updated, and data sets are constantly modified. CDFW may provide data upon request, but, unless otherwise stated, does not endorse any particular analytical methods, interpretations, or conclusions based upon the data it provides. We also thank Kelly Spaulding in the National Marine Fisheries Service Office of Law Enforcement for providing access to Vessel Monitoring System Data.”

“This work was funded by the US Department of the Interior, Bureau of Ocean Energy Management (BOEM), Environmental Studies Program, Washington, DC (Agreement Number #M16AC00023, https://www.boem.gov/) and the California Ocean Protection Council (OPC) (Agreement #C0210403, https://www.opc.ca.gov/) to BR, CW, and RW. BOEM and OPC had a role in reviewing the manuscript. FP at BOEM assisted in manuscript preparation.”

4. We note that Figures 2a,2b,3,4,5,8,9,10b and S5-S25 Fig. in your submission contain [map/satellite] images which may be copyrighted. All PLOS content is published under the Creative Commons Attribution License (CC BY 4.0), which means that the manuscript, images, and Supporting Information files will be freely available online, and any third party is permitted to access, download, copy, distribute, and use these materials in any way, even commercially, with proper attribution. For these reasons, we cannot publish previously copyrighted maps or satellite images created using proprietary data, such as Google software (Google Maps, Street View, and Earth). For more information, see our copyright guidelines: http://journals.plos.org/plosone/s/licenses-and-copyright.

a. You may seek permission from the original copyright holder of Figures 2a,2b,3,4,5,8,9,10b and S5-S25 Fig. to publish the content specifically under the CC BY 4.0 license. 

Additional Editor Comments:

#Editor Comment:

The methodologies employed in this study adhere to conventional standards. Notably, these methodologies were initially introduced and documented in a publication dating back a minimum of 13 years, as evidenced by the referenced paper below.

Gerritsen H, Lordan C. Integrating vessel monitoring systems (VMS) data with daily

catch data from logbooks to explore the spatial distribution of catch and effort at high

resolution. ICES J Mar Sci. 2011 Jan 1;68(1):245–52.

In light of this, I kindly request that the authors furnish additional elucidation regarding the distinctive attributes of their approach in comparison to antecedent work in the field. Such clarification will be instrumental in ascertaining the novel contributions and unique merits of their research.

#Reviewer 1

Main Comments

The results from this paper have substantial utility for management/marine planning. My main thoughts are for protected species (ESA listed species) that overlap with these fisheries but as the authors highlight, there is utility for stock assessment as well and for marine spatial planning with the future uses of the marine environment (wind power, aquaculture, etc). Therefore this paper is very relevant and important for management and policy.

One main concern is how comprehensive the data/analysis is for fisheries other than groundfish and how certain statements are confusing – some statements make it sounds like ALL commercial fisheries are fully covered while others make it sound like only groundfish fisheries are fully represented. The paper presents results for other fisheries such as HMS and salmon, but is unclear how representative the data is for these given the management mandates for the use of VMS are mainly only for groundfish. The authors say on lines 141-145 – “In addition, vessels that operate under a permit associated with VMS (e.g., Pacific Coast groundfish limited entry permit) are also required to operate a VMS transceiver during all fishing trips, even when fishing with gear or targeting fish species outside the jurisdiction of the VMS requirement (e.g., Dungeness crabfishery, salmon fishery).” So at least some of the Dungeness and salmon effort is represented but how much? Any way to quantify? Also, the authors state on lines 136-137 – “VMS is required on drift gillnet vessels participating in Highly Migratory Species (HMS) fisheries,” – what percent then of all HMS fishing is done by drift gillnet? Does this data represent the majority of HMS fishing or does a substantial amount use a different gear?

Lines 103-104 in the introduction say that spatial maps were produced for “all commercial fisheries” but is that the case if VMS data is not mandated for all commercial fisheries like salmon? Or am I missing something?

Lines 147-149 say - “Thus, VMS not only has nearly complete coverage of commercial groundfish fisheries operating in federal waters, but also covers many other commercial fisheries in the region with varying levels of representation.” – but to what degree of representation is important. Some of this is noted on lines 287-288 for sea cucumber and ridgeback prawns, but what about other non-groundfish species?

Wondering then if the title should be more explicit that the majority of the research and data are for groundfish? Maybe a sub-title specifying the focus on groundfish?

Also, VMS data used was from 2010-2017 – is there not more recent data? How easily can the analysis be updated to incorporate more recent data as it becomes available?

And, that year set used incorporates certain “blob” heatwave years. Did fishing vessel use of the marine environment shift during those heat wave years? Lines 527-532 in the results indicate that there has been shifts overtime, so any specific different associated with blob years?

Minor Comments

Methods

Lines 166-171 – Remind the reader why you are looking at fish ticket data as well and why only for California?

Lines 201- 204 – how often where there was a 5+ hour gap in coverage? What percent of the data? Is there any concern that for some reason VMS was turned off or there was a loss in transmission?

Lines 238-240 – why replace null values and/or $0/lb values with the median non $0 price? Why do certain data get entered as null or 0 in the fish ticket database? Is there a reason for the data being entered as such that would suggest it’s a true 0? Possibly it’s discard/bycatch species that don’t get sold or retained?

Line 297, Section 2.4 – is there a reason to talk about fish ticket data before this section? The order is a bit confusing to talk about fish ticket data before this section if only used for the analysis talked about in section 2.4.

Lines 319-327 – so it’s assumed that fishing effort and catch is equal across the whole track? How likely is that assumption?

Results

Lines 456-457 – Remind reader main reason why unable to match the data between fish tickets and VMS. Is this because a lot of the VMS data is coming from Oregon/Washington?

Discussion

Lines 574-577 – Similar to questions above, so what percent of the Dungeness crab fishery is represented by this data? Anyway to estimate?

Lines 631- 634 – How do you know the relative spatial patterns are correctly represented? What percent of the other declaration codes are represented by this data?

Figures

These are somewhat pixelated and hard to see detail – possibly an artifact of the creation of the pdf? But higher resolution would be great.

#Reviewer 2

This paper used VMS and fisheries landings to map fishing effort, and to spatially allocate catch value in Californian waters. VMS data were also examined more widely across the US West Coast.

It was very well written and very relevant for management of the marine environment in US West Coast waters.

The methods used are largely based on published approaches, and validation was built into the approach (e.g. comparison against fisheries observer program data, correlations test and alternative rules).

Assumptions and limitations are already addressed.

My minor comment is to explain better why the minimum slope of the histogram between 2 and 6 knots was used an alternative approach – is there a reference for this?

Reviewers' comments:

Reviewer's Responses to Questions

**Comments to the Author**

1. Is the manuscript technically sound, and do the data support the conclusions?

Reviewer #1: Yes

Reviewer #2: Yes

2. Has the statistical analysis been performed appropriately and rigorously? 

Reviewer #1: Yes

Reviewer #2: Yes

3. Have the authors made all data underlying the findings in their manuscript fully available?

Reviewer #1: No

Reviewer #2: No

4. Is the manuscript presented in an intelligible fashion and written in standard English?

Reviewer #1: Yes

Reviewer #2: Yes

5. Review Comments to the Author

Reviewer #1: Main Comments

The results from this paper have substantial utility for management/marine planning. My main thoughts are for protected species (ESA listed species) that overlap with these fisheries but as the authors highlight, there is utility for stock assessment as well and for marine spatial planning with the future uses of the marine environment (wind power, aquaculture, etc). Therefore this paper is very relevant and important for management and policy.

One main concern is how comprehensive the data/analysis is for fisheries other than groundfish and how certain statements are confusing – some statements make it sounds like ALL commercial fisheries are fully covered while others make it sound like only groundfish fisheries are fully represented. The paper presents results for other fisheries such as HMS and salmon, but is unclear how representative the data is for these given the management mandates for the use of VMS are mainly only for groundfish. The authors say on lines 141-145 – “In addition, vessels that operate under a permit associated with VMS (e.g., Pacific Coast groundfish limited entry permit) are also required to operate a VMS transceiver during all fishing trips, even when fishing with gear or targeting fish species outside the jurisdiction of the VMS requirement (e.g., Dungeness crabfishery, salmon fishery).” So at least some of the Dungeness and salmon effort is represented but how much? Any way to quantify? Also, the authors state on lines 136-137 – “VMS is required on drift gillnet vessels participating in Highly Migratory Species (HMS) fisheries,” – what percent then of all HMS fishing is done by drift gillnet? Does this data represent the majority of HMS fishing or does a substantial amount use a different gear?

Lines 103-104 in the introduction say that spatial maps were produced for “all commercial fisheries” but is that the case if VMS data is not mandated for all commercial fisheries like salmon? Or am I missing something?

Lines 147-149 say - “Thus, VMS not only has nearly complete coverage of commercial groundfish fisheries operating in federal waters, but also covers many other commercial fisheries in the region with varying levels of representation.” – but to what degree of representation is important. Some of this is noted on lines 287-288 for sea cucumber and ridgeback prawns, but what about other non-groundfish species?

Wondering then if the title should be more explicit that the majority of the research and data are for groundfish? Maybe a sub-title specifying the focus on groundfish?

Also, VMS data used was from 2010-2017 – is there not more recent data? How easily can the analysis be updated to incorporate more recent data as it becomes available?

And, that year set used incorporates certain “blob” heatwave years. Did fishing vessel use of the marine environment shift during those heat wave years? Lines 527-532 in the results indicate that there has been shifts overtime, so any specific different associated with blob years?

Minor Comments

Methods

Lines 166-171 – Remind the reader why you are looking at fish ticket data as well and why only for California?

Lines 201- 204 – how often where there was a 5+ hour gap in coverage? What percent of the data? Is there any concern that for some reason VMS was turned off or there was a loss in transmission?

Lines 238-240 – why replace null values and/or $0/lb values with the median non $0 price? Why do certain data get entered as null or 0 in the fish ticket database? Is there a reason for the data being entered as such that would suggest it’s a true 0? Possibly it’s discard/bycatch species that don’t get sold or retained?

Line 297, Section 2.4 – is there a reason to talk about fish ticket data before this section? The order is a bit confusing to talk about fish ticket data before this section if only used for the analysis talked about in section 2.4.

Lines 319-327 – so it’s assumed that fishing effort and catch is equal across the whole track? How likely is that assumption?

Results

Lines 456-457 – Remind reader main reason why unable to match the data between fish tickets and VMS. Is this because a lot of the VMS data is coming from Oregon/Washington?

Discussion

Lines 574-577 – Similar to questions above, so what percent of the Dungeness crab fishery is represented by this data? Anyway to estimate?

Lines 631- 634 – How do you know the relative spatial patterns are correctly represented? What percent of the other declaration codes are represented by this data?

Figures

These are somewhat pixelated and hard to see detail – possibly an artifact of the creation of the pdf? But higher resolution would be great.

Reviewer #2: This paper used VMS and fisheries landings to map fishing effort, and to spatially allocate catch value in Californian waters. VMS data were also examined more widely across the US West Coast.

It was very well written and very relevant for management of the marine environment in US West Coast waters.

The methods used are largely based on published approaches, and validation was built into the approach (e.g. comparison against fisheries observer program data, correlations test and alternative rules).

Assumptions and limitations are already addressed.

My minor comment is to explain better why the minimum slope of the histogram between 2 and 6 knots was used an alternative approach – is there a reference for this?

6. PLOS authors have the option to publish the peer review history of their article (what does this mean?). If published, this will include your full peer review and any attached files.

Reviewer #1: No

Reviewer #2: No

---

## [Author Response · Author response to Decision Letter 0]

11 Dec 2023

Response to Reviewers for PONE-D-23-21225

High resolution assessment of commercial fisheries activity along the US West Coast using Vessel Monitoring System data with a case study using California groundfish fisheries

Yi-Hui Wang, Benjamin I. Ruttenberg, Ryan K. Walter, Frank Pendleton, Jameal Samhouri, Owen Liu, Crow White

We would like to thank the Editor and Reviewers for their time and feedback. A point-by-point response is given below the Reviewer’s comments in black and our responses in blue (NOTE: original document shows responses in blue).

“The views and conclusions contained in this document are those of the authors and should not be interpreted as representing the opinions or policies of the U.S. Government. Mention of trade names or commercial products does not constitute their endorsement by the U.S. Government. Study collaboration and funding were provided by the U.S. Department of the Interior, Bureau of Ocean Energy Management, Environmental Studies Program, Washington, DC under Agreement Number Ml6AC0002. Additional funding was provided the California Ocean Protection Council (OPC), Agreement Number C0210403. We are indebted to Brian Owens, Todd Neahr and Paulo Serpa at the California Department of Fish and Wildlife (CDFW) for providing access to and sharing insights about CDFW landings data. CDFW acquires data from its own fisheries management activities and from mandatory reporting requirements on the commercial and recreational fishery pursuant to the Fish and Game Code and the California Code of Regulations. These data are constantly being updated, and data sets are constantly modified. CDFW may provide data upon request, but, unless otherwise stated, does not endorse any particular analytical methods, interpretations, or conclusions based upon the data it provides. We also thank Kelly Spaulding in the National Marine Fisheries Service Office of Law Enforcement for providing access to Vessel Monitoring System Data.”

“This work was funded by the US Department of the Interior, Bureau of Ocean Energy Management (BOEM), Environmental Studies Program, Washington, DC (Agreement Number #M16AC00023, https://www.boem.gov/) and the California Ocean Protection Council (OPC) (Agreement #C0210403, https://www.opc.ca.gov/) to BR, CW, and RW. BOEM and OPC had a role in reviewing the manuscript. FP at BOEM assisted in manuscript preparation.”

The funding-related text has been removed from the manuscript. There is no change in our Funding Statement.

There is no change in our Data Availability statement.

4. We note that Figures 2a,2b,3,4,5,8,9,10b and S5-S25 Fig. in your submission contain [map/satellite] images which may be copyrighted. All PLOS content is published under the Creative Commons Attribution License (CC BY 4.0), which means that the manuscript, images, and Supporting Information files will be freely available online, and any third party is permitted to access, download, copy, distribute, and use these materials in any way, even commercially, with proper attribution. For these reasons, we cannot publish previously copyrighted maps or satellite images created using proprietary data, such as Google software (Google Maps, Street View, and Earth). For more information, see our copyright guidelines: http://journals.plos.org/plosone/s/licenses-and-copyright.

a. You may seek permission from the original copyright holder of Figures 2a,2b,3,4,5,8,9,10b and S5-S25 Fig. to publish the content specifically under the CC BY 4.0 license. 

The authors have confirmed with the journal office that these images were flagged in error. Still, we have confirmed that all background maps are either in the public domain or open source requiring no permission. To recognize the copyright for the background maps, we have credited the sources for background maps in the caption of Figure 2, which is the first map displayed in the manuscript. Additionally, we modified the rightmost map in Figure 3 so it now uses a map in the public domain from OpenStreetMap as the background map.

Additional Editor Comments:

#Editor Comment:

The methodologies employed in this study adhere to conventional standards. Notably, these methodologies were initially introduced and documented in a publication dating back a minimum of 13 years, as evidenced by the referenced paper below.

Gerritsen H, Lordan C. Integrating vessel monitoring systems (VMS) data with daily

catch data from logbooks to explore the spatial distribution of catch and effort at high

resolution. ICES J Mar Sci. 2011 Jan 1;68(1):245–52.

In light of this, I kindly request that the authors furnish additional elucidation regarding the distinctive attributes of their approach in comparison to antecedent work in the field. Such clarification will be instrumental in ascertaining the novel contributions and unique merits of their research.

While our analysis of VMS data to quantify fisheries activities builds upon and slightly advances the methods by previous studies (e.g., we matched VMS data to CDFW fish ticket data, while Gerritsen and Lordan 2011 matched VMS to logbook data; and we used vessel speed histogram data to determine fishing vs non-fishing activity for individual declaration codes, as opposed to single minimum and maximum cut-off values used by Gerritsen and Lordan 2011), the main novelty of our study is the application of these methods to a broad range of US West Coast fisheries (and the California groundfish fisheries only for the matched VMS-fish ticket data analysis). We have modified the end of the Introduction to better acknowledge previous research in relation to this study. 

#Reviewer 1

Main Comments

The results from this paper have substantial utility for management/marine planning. My main thoughts are for protected species (ESA listed species) that overlap with these fisheries but as the authors highlight, there is utility for stock assessment as well and for marine spatial planning with the future uses of the marine environment (wind power, aquaculture, etc). Therefore this paper is very relevant and important for management and policy.

One main concern is how comprehensive the data/analysis is for fisheries other than groundfish and how certain statements are confusing – some statements make it sounds like ALL commercial fisheries are fully covered while others make it sound like only groundfish fisheries are fully represented. The paper presents results for other fisheries such as HMS and salmon, but is unclear how representative the data is for these given the management mandates for the use of VMS are mainly only for groundfish. The authors say on lines 141-145 – “In addition, vessels that operate under a permit associated with VMS (e.g., Pacific Coast groundfish limited entry permit) are also required to operate a VMS transceiver during all fishing trips, even when fishing with gear or targeting fish species outside the jurisdiction of the VMS requirement (e.g., Dungeness crab fishery, salmon fishery).” So at least some of the Dungeness and salmon effort is represented but how much? Any way to quantify? Also, the authors state on lines 136-137 – “VMS is required on drift gillnet vessels participating in Highly Migratory Species (HMS) fisheries,” – what percent then of all HMS fishing is done by drift gillnet? Does this data represent the majority of HMS fishing or does a substantial amount use a different gear?

We have clarified text in the manuscript to explain what fisheries are represented by each component of our analysis. There are few available sources to quantify coverage rate of the VMS data for US West Coast fisheries (except groundfish, which is presumed to be >99%; but see comment below for Dungeness crab, Liu et al. 2023). Consequently, beyond the maps we have generated (indicating relative effort across space), we are not able to provide quantitative representations of fishing effort (e.g., absolute total effort) for the fisheries that are not required to always use VMS (like for the groundfish fisheries). Additionally, we are unable to obtain the percentage estimate of HMS fishing with different gears.

Lines 103-104 in the introduction say that spatial maps were produced for “all commercial fisheries” but is that the case if VMS data is not mandated for all commercial fisheries like salmon? Or am I missing something?

The details of VMS data are provided in the Data and Method section (specifically Section 2.1). To avoid any confusion in the Introduction section, we have modified this sentence to read “We processed the data, and produced spatial maps of relative fishing effort across the entire US West Coast, encompassing a variety of commercial fisheries that collected VMS data (see Data and Methods).”

Lines 147-149 say - “Thus, VMS not only has nearly complete coverage of commercial groundfish fisheries operating in federal waters, but also covers many other commercial fisheries in the region with varying levels of representation.” – but to what degree of representation is important. Some of this is noted on lines 287-288 for sea cucumber and ridgeback prawns, but what about other non-groundfish species?

We appreciate the Reviewer bringing this to our attention. We have modified the text to clarify that VMS is required for groundfish fisheries operating in federal waters but only for other fisheries if vessels fish for federally-managed groundfish during the year. We have also modified the text in the Discussion to highlight the fact that the spatial patterns for Dungeness crab data from VMS data (for which VMS is not required) matched those from Oregon Department of Fish and Wildlife, suggesting that VMS data may capture spatial fishing activity for fisheries other than groundfish as well.

Wondering then if the title should be more explicit that the majority of the research and data are for groundfish? Maybe a sub-title specifying the focus on groundfish?

We appreciate Reviewer’s suggestion and have added a subtitle that specifically addresses our analysis on groundfish. The revised title now reads as “High resolution assessment of commercial fisheries activity along the US West Coast using Vessel Monitoring System data with California groundfish fisheries case study”.

Also, VMS data used was from 2010-2017 – is there not more recent data? How easily can the analysis be updated to incorporate more recent data as it becomes available?

And, that year set used incorporates certain “blob” heatwave years. Did fishing vessel use of the marine environment shift during those heat wave years? Lines 527-532 in the results indicate that there has been shifts overtime, so any specific different associated with blob years?

We originally obtained VMS data for 2010-2017 for this study. While more recent data may be available, the time and resources required to obtain, process and analyze additional fishery data is beyond the scope of this study. The analysis of additional years data is left for future studies. 

While we agree that the Reviewer’s question about the impact of MHWs on fishing activity is very interesting, untangling the connection between changes in fishing activities and extreme heatwave events is beyond the scope of this study (and would also require additional data on spatial temperature anomalies). Unfortunately, we are unable to address it in this paper.

Minor Comments

Methods

Lines 166-171 – Remind the reader why you are looking at fish ticket data as well and why only for California?

Fish ticket data provides landings and ex-vessel values – data not recorded by VMS data. Thus, combining the Fish ticket data with the VMS tracking data enables high resolution estimation of fisheries landings and value. We were able to obtain fish landings data (“fish tickets”) from the California Department of Fish and Wildlife (CDFW), but not for other states. As a result, we were only able to explore patterns of landings and value in California. We have modified the text to: “This study used commercial landings receipts, known as ‘fish tickets’, which offer information about landings and ex-vess

---

## [Decision Letter · Decision Letter 1]

1 Feb 2024

High resolution assessment of commercial fisheries activity along the US West Coast using Vessel Monitoring System data with a case study using California groundfish fisheries

PONE-D-23-21225R1

Dear Dr. Ruttenberg,

We’re pleased to inform you that your manuscript has been judged scientifically suitable for publication and will be formally accepted for publication once it meets all outstanding technical requirements.

Kind regards,

Md. Naimur Rahman

Academic Editor

PLOS ONE

Additional Editor Comments (optional):

Reviewers' comments:

Reviewer's Responses to Questions

**Comments to the Author**

1. If the authors have adequately addressed your comments raised in a previous round of review and you feel that this manuscript is now acceptable for publication, you may indicate that here to bypass the “Comments to the Author” section, enter your conflict of interest statement in the “Confidential to Editor” section, and submit your "Accept" recommendation.

Reviewer #1: All comments have been addressed

Reviewer #2: All comments have been addressed

2. Is the manuscript technically sound, and do the data support the conclusions?

Reviewer #1: Yes

Reviewer #2: Yes

3. Has the statistical analysis been performed appropriately and rigorously? 

Reviewer #1: Yes

Reviewer #2: Yes

4. Have the authors made all data underlying the findings in their manuscript fully available?

Reviewer #1: No

Reviewer #2: Yes

5. Is the manuscript presented in an intelligible fashion and written in standard English?

Reviewer #1: Yes

Reviewer #2: Yes

6. Review Comments to the Author

Reviewer #1: I really appreciate the thorough methods - easy to follow what was done. My one last comment is if these methods and/or products are meant to be used by managers (stated in the conclusion), then how will they be available and how will they be updated to represent more recent data?

Reviewer #2: I think that the authors have addressed reviewers comments adequately, and I have no further comments.

7. PLOS authors have the option to publish the peer review history of their article (what does this mean?). If published, this will include your full peer review and any attached files.

Reviewer #1: **Yes: **Laura Koehn

Reviewer #2: No

---

## [Editor Report · Acceptance letter]

11 May 2024

PONE-D-23-21225R1 

PLOS ONE

Dear Dr. Ruttenberg, 

I'm pleased to inform you that your manuscript has been deemed suitable for publication in PLOS ONE. Congratulations! Your manuscript is now being handed over to our production team.

Kind regards, 

on behalf of

Mr Md. Naimur Rahman 

Academic Editor

PLOS ONE